# Feature Representation Transferring to Lightweight Models via *Perception Coherence*

**Hai-Vy Nguyen**                                    *hai-vy.nguyen@ampere.cars*
*Ampere Software Technology – Institut de mathématiques de Toulouse*
*– Institut de Recherche en Informatique de Toulouse*

**Fabrice Gamboa**                          *fabrice.gamboa@math.univ-toulouse.fr*
*Institut de mathématiques de Toulouse*

**Sixin Zhang**                                          *sixin.zhang@irit.fr*
*Institut de Recherche en Informatique de Toulouse*

**Reda Chhaibi**                               *reda.chhaibi@univ-cotedazur.fr*
*Laboratoire Jean Alexandre Dieudonné, Université Côte d'Azur*

**Serge Gratton**                                 *serge.gratton@toulouse-inp.fr*
*Institut de Recherche en Informatique de Toulouse*

**Thierry Giaccone**                            *thierry.giaccone@ampere.cars*
*Ampere Software Technology*

**Reviewed on OpenReview:** *https://openreview.net/forum?id=yQbNbeSEUq*

## Abstract

In this paper, we propose a method for transferring feature representation to lightweight student models from larger teacher models. We mathematically define a new notion called *perception coherence*. Based on this notion, we propose a loss function, which takes into account the dissimilarities between data points in feature space through their ranking. At a high level, by minimizing this loss function, the student model learns to mimic how the teacher model *perceives* inputs. More precisely, our method is motivated by the fact that the representational capacity of the student model is weaker than the teacher model. Hence, we aim to develop a conceptually new method allowing for a better relaxation. This means that, the student model does not need to preserve the absolute geometry of the teacher one, while preserving global coherence through dissimilarity ranking. Importantly, while rankings are defined only on finite sets, our notion of *perception coherence* extends them into a probabilistic form. This formulation depends on the input distribution and applies to general dissimilarity metrics. Our theoretical insights provide a probabilistic perspective on the process of feature representation transfer. Our experimental results show that our method outperforms or achieves on-par performance with strong baseline methods for representation transfer, particularly class-unaware ones.

## 1 Introduction

**The need of lightweight models and Knowledge distilling (KD).** Deep learning models are at the forefront of performance in different domains and tasks, such as classification (He et al., 2016; Tan & Le, 2019; 2021; He et al., 2020) or object detection (Ren et al., 2015; Lin et al., 2017a;b; He et al., 2020). However, as the performance of neural networks increases, so does the cost of increasingly large models. In many applications where resources are limited (e.g. mobile devices) or one needs a fast execution, it is preferable to use lightweight models. Among the many approaches to obtain lightweight models, KD (Buciluǎ et al., 2006; Hinton et al., 2015; Wang & Yoon, 2021) stands out as a particularly interesting direction. A KD setup essentially consists of a *teacher model* and a *student model*. The student model (here the targeted lightweight model) is generally smaller than the teacher model, and therefore more efficient for the task in question. At a high level, the KD process consists in teaching the student model so that it mimics the way

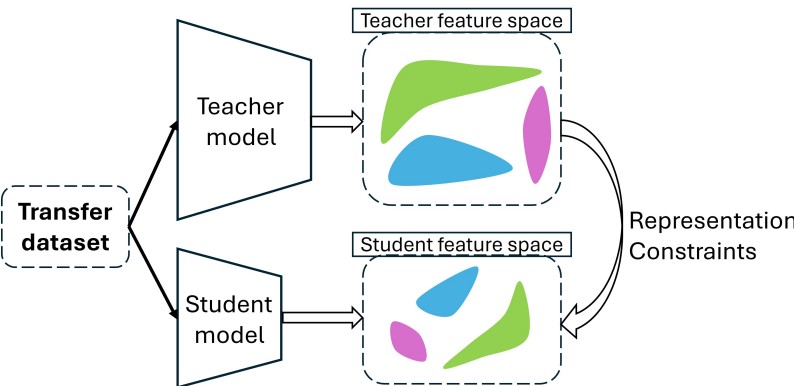

Figure 1: Illustration of feature representation transfer. A non-labeled transfer dataset is passed through both teacher and student models to obtain their respective feature representations. The transfer process consists in training the student model to somehow capture the input–input relationships encoded in the teacher model's feature space.

the teacher model understands the input. This is usually achieved by using a so-called *transfer dataset*. The student model is trained to behave like the teacher model on this set in some way . Many previous works have shown that KD improves the generalization capacity of the student model (Hinton et al., 2015; Chen et al., 2017; Zhu et al., 2018; Beyer et al., 2022; Huang et al., 2022). Different up-to-date KD approaches will be discussed in Section 2.

**Feature representation transferring.** Classical KD techniques generally focus on matching the outputs of student and teacher models, typically for classification tasks (Hinton et al., 2015). However, this approach has limitations, such as requiring the same number of classes between the teacher and student models. To address these issues, previous works (Passalis & Tefas, 2018a; Passalis et al., 2020b; Tian et al., 2020; Park et al., 2019) propose to directly transfer the feature space representations through learning. This enables the student model to better capture the input-input relations represented in the feature space of the teacher model, by preserving its geometry and distribution (illustrated in Fig. 1). This leads to more effective knowledge transfer. Furthermore, note that this approach is an unsupervised transferring method, as it only uses a transfer set without labels. As a consequence, the learned features can be used for various downstream tasks. Because of these appealing properties, our method aligns with this last approach.

**Motivation and an overview of our method.** Generally, the student model is much smaller than the teacher model. Thus, in terms of representational power, the student model cannot learn to produce features with the exact same geometry as the teacher model. Hence, we aim to develop a method such that (1) the student model does not need to copy the feature representation of the teacher model; (2) the learned representation retains a certain overall consistency with the teacher model's feature representation and (3) the method works for representation transferring between spaces having different dimensions. For these purposes, we develop a novel fairly simple notion called *perception coherence*, that is explained subsequently. To begin with, let us consider a reference input $x$ and a set of $n$ point $\{x_i\}_i^n$ (called *compared set*). These points are fed into the student and teacher models and are mapped into the feature space of each model. The feature space reflects how the models *perceive* the inputs. Placing ourselves in the two feature spaces, let $d_1$ and $d_2$ be dissimilarity measures for the teacher and student models, respectively. Then, if $d_1(x, x_i) \leq d_1(x, x_j)$ for some $i, j \in \{1, 2, \ldots, n\}$ $(i \neq j)$, we expect that $d_2(x, x_i) \leq d_2(x, x_j)$ also holds. Less formally, if the teacher model *perceives* $x$ to be more similar to $x_i$ than to $x_j$, we expect the student to have the same perception. We refer to this as the property of *perception coherence*. By adopting this behavior for different reference points and compared sets, the student model learns to have a coherent perception with respect to the teacher model. Moreover, since such rankings are only defined over finite sets, we further generalize this concept into a probabilistic form—our formal definition of *perception coherence* (Definition 3.2)—which depends on the input distribution and applies to general dissimilarity metrics.

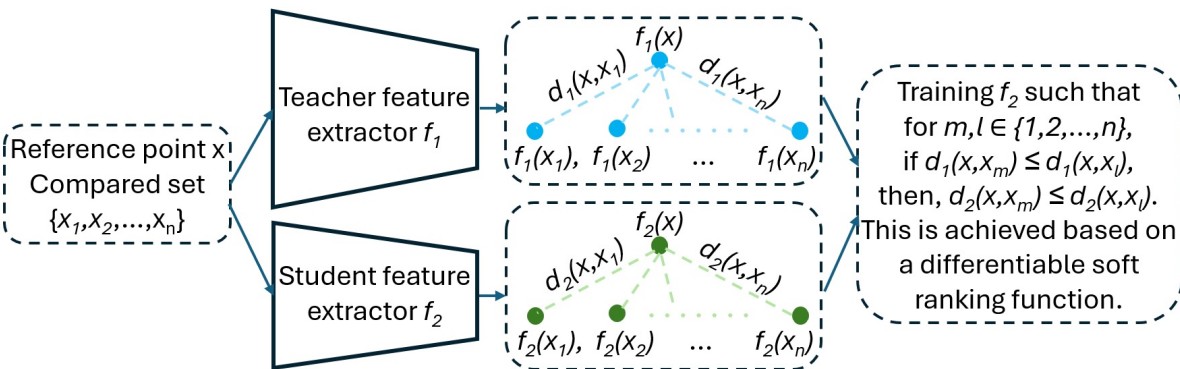

Figure 2: General scheme of our method. In each mini-batch, we use each input as the reference point to compute the dissimilarity to the others (compared set). Then, using loss function in Eq. (4), the student model is trained to respect the perception coherence. Notice that $f_1$ and $f_2$ can be the whole model (when using the output layer) or a part of the model (when using an intermediate layer).

The general idea of our method is illustrated in Figure 2 (further details will be illustrated in Figure 3). Note that we do not quantify *how much* more similar $x_i$ is to $x$ than $x_j$. Instead, we only preserve the relative ranking of dissimilarities. This relaxation provides flexibility, as the student model is not required to replicate the exact geometry or distribution of the teacher model's feature space. However, the ranking operation is non-differentiable. To address this, we introduce a simple differentiable approximation based on a soft ranking function (Eq. (3)). Our contribution can be summarized as follows:

- We propose a new method for distilling knowledge to lightweight models, based on deep representation transferring. Central to our approach is a novel probabilistic concept, *perception coherence*, which naturally leads to a simple and easy-to-implement loss function (Eq. (4)) based on a soft ranking operation.

- We provide theoretical insights through a probabilistic framework that connects representation transfer to local and global dissimilarity rankings in the feature spaces of teacher and student models. In particular, we show in Section 4.1 that the *perception coherence level* can be approximated via ranking operations on mini-batches with convergence rate $O(1/\sqrt{B})$, where $B$ is the mini-batch size. Furthermore, we shall also demonstrate that a larger *global expected coherence level* leads to a higher chance that the learned representation respects the dissimilarity ranking of the teacher one (Sections 4.2 and 4.3).

- We conduct experiments on retrieval and classification tasks to compare our method with strong baseline methods. The results show that our method outperforms or achieves competitive performance with strong baseline methods on lightweight representation transferring settings. Our code is available at perception-coherence-code.

## 2   Related Work

**KD based on soft label produced by teacher model.** This approach has been studied since the early days of neural networks (Buciluă et al., 2006). Therein, the authors show that knowledge can be transferred from a large ensemble of models to a single small model. A standard KD method consists in using output of classification models (Hinton et al., 2015; Tang et al., 2016; Huang et al., 2022), with a temperature, leading to smoother soft labels. Based on the same spirit, (Chan et al., 2015) proposes to match the output distributions using Kullback-Leibler divergence. Soft label was also used in the context of domain adaption (Tzeng et al., 2015). While all these methods show the benefits of soft labels, the main limit of this approach is the need of having the same number of classes between models.

**KD based on distance matching.** In Romero et al. (2014), the authors propose to minimize the distance between student and teacher's features to guide the student model during training. However, in the case of dimension mismatch between the models, one need to pass through a linear transformation, which leads to

substantial information loss. In Yim et al. (2017), the authors propose to minimize distance of the so-call *FSP matrix* between teacher and student models (capturing inter-layer relations). More advanced methods consist in minimizing the distance between the derivatives w.r.t. to the features (Czarnecki et al., 2017) or between the attention map of student and teacher (Zagoruyko & Komodakis, 2016). All these approaches require the dimension match, and need some form of interpolation in the case of dimension mismatch. Overcoming the drawback of dimension mismatch, Passalis & Tefas (2018b) proposes to match the the pair-wise similarity matrix of the training points (in the transfer set) between the student and teacher models. The method proposed in Yu et al. (2019) is to match the distance between pairs of samples. However, this brute force approach is not likely to work as the student model is much smaller than the teacher one (as discussed in Passalis et al. (2020b)).

**Feature representation transfer.** Some previous works treat the representation transfer as the problem of distribution matching, based on Maximum Mean Discrepancy (Huang & Wang, 2017) or adversarial training (Belagiannis et al., 2018). The latter method is inspired by Generative Adversarial Networks (GANs) Goodfellow et al. (2014). One major drawback of distribution matching is the need of having the same dimension between the two feature spaces. Probabilistic frameworks based on mutual information are proposed in Ahn et al. (2019); Tian et al. (2020). However, the method of Ahn et al. (2019) requires a Gaussian assumption (this is not necessarily satisfied in practice) while the method of Tian et al. (2020) requires the input features to have the same dimension (hence requiring a linear transformation, which may reduce the useful information). Overcoming the requirement of dimension match, Park et al. (2019) proposes to transfer the input relations using distance-wise and angle-wise distillation loss. However, the structural information is quite limited, as it only considers a single triple of points at a time. Overcoming this drawback, Passalis & Tefas (2018a) proposes a method of probabilistic knowledge transfer (PKT). The high level idea of this last method is to transfer the local geometry learned by the teacher model based on kernel techniques. Based on this work, Passalis et al. (2020a;b) basically share the same idea, but they apply the technique for multiples intermediate layers with an auxiliary model. The benefits of using auxiliary models are discussed in Mirzadeh et al. (2020). As in Passalis & Tefas (2018a); Passalis et al. (2020a;b), our method takes into account the relation between the inputs and allows representation transfer between space having different dimensions. However, in Passalis & Tefas (2018a); Passalis et al. (2020a;b), a common kernel is used for all points (e.g. a Gaussian one with a fixed bandwidth), which cannot adapt to local density variation. This contrasts with our method, where we only rank the dissimilarities but do not use the dissimilarity magnitude. Thus, by using the dissimilarity rankings (or equivalently *cumulative function of dissimilarities*), our method captures local probabilistic closeness — a property that raw distance magnitudes fail to reflect in the presence of density variations.

**Class-awareness.** Knowledge distillation is a broad research domain, known for leveraging the "knowledge" of a teacher model to improve a student model. However, most state-of-the-art methods adopt a *class-aware* approach. This means that these methods rely on class information in some form—most notably, the number of classes. The standard approach involves matching the output logits (Hinton et al., 2015; Huang et al., 2022). More recent class-aware methods reuse the teacher's classifier (Chen et al., 2022) and then retrain the student encoder. Other methods employ the teacher's classification head to implicitly align the student's intermediate representations with those of the teacher (Wang et al., 2024). While such methods demonstrate strong performance on certain benchmarks, they are often tailored to specific tasks. In contrast, our method is conceptually designed for *general-purpose representation transfer*, applicable to any feature space equipped with a pairwise dissimilarity measure. This *class-unaware* formulation conceptually enables broader applicability, e.g., regression models or even hand-crafted features.

## 3 Our method

### 3.1 Perception coherence and other notions

We consider a task where the input space is denoted by $\mathcal{X}$ associated with the underlying distribution $\mathcal{D}_{\mathcal{X}}$. Let us consider two feature extractors $f_1 : \mathcal{X} \mapsto \mathcal{F}_1$ and $f_2 : \mathcal{X} \mapsto \mathcal{F}_2$, endowed with symmetric dissimilarity metrics $d_{f_1} : \mathcal{F}_1 \times \mathcal{F}_1 \mapsto \mathbb{R}$ and $d_{f_2} : \mathcal{F}_2 \times \mathcal{F}_2 \mapsto \mathbb{R}$, respectively. For ease of following, we can think of $f_1$ and $f_2$

as teacher and student models, respectively. For brevity, for any $x, x' \in \mathcal{X}$, let $d_1(x, x') = d_{f_1}(f_1(x), f_1(x'))$ and $d_2(x, x') = d_{f_2}(f_2(x), f_2(x'))$.

At a high level, if $f_1$ *perceives* $x$ to be more similar to $x_1$ than to $x_2$, then it should be also the case for $f_2$ after the representation transferring. This is represented through the feature space of each model, endowed with their corresponding dissimilarity metric. We present this intuition via a more formal concept named *absolute perception coherence* as follows.

**Definition 3.1 (Absolute perception coherence)** *We say that $f_2$ is absolutely perception coherent with $f_1$ at $x \in \mathcal{X}$ if $d_1(x, x_2) \geq d_1(x, x_1)) \Leftrightarrow d_2(x, x_2) \geq d_2(x, x_1)$, for any $x_1, x_2 \in \mathcal{X}$.*

However, the absolute perception coherence is too strict. Thus, we shall introduce a more relaxed notion subsequently, based on a probabilistic framework by using the distribution $\mathcal{D}_{\mathcal{X}}$. To begin with, given $x, x' \in \mathcal{X}$, let us define the cumulative functions as follows

$$F_1(x, x') := \mathbb{P}_X \left( d_1(x, X) \leq d_1(x, x') \right), \ F_2(x, x') := \mathbb{P}_X \left( d_2(x, X) \leq d_2(x, x') \right). \tag{1}$$

**Remark 3.1** *We have the following remarks on the relation between the previous function and the ordering of dissimilarities to a given reference point $x$.*

- *$F_i(x, x_1) \leq F_i(x, x_2) \Leftrightarrow d_i(x, x_1) \leq d_i(x, x_2)$, for $i \in \{1, 2\}$.*

- *A small value of $F_i(x, x')$ means that $x'$ is probabilistically close to $x$. In other words, if we choose randomly another point $\widetilde{x}$ according to $\mathcal{D}_{\mathcal{X}}$, there is a high chance that $d_i(x, \widetilde{x}) \geq d_i(x, x')$.*

- *Similarly, a large value of $F_i(x, x')$ means that $x'$ is probabilistically far from $x$.*

- *Thus, for each model $f_i$, $F_i(x, x')$ represents "probabilistic distance" between $x$ and $x'$.*

**Remark 3.2 (Relation between cumulative function and ranking)** *Let us consider a discrete distribution uniformly distributed over $N$ examples $\{x_i\}_{i=1}^N$ and the extractor $f$ endowed with a dissimilarity metric $d$. We consider $x_i$ as the reference point and we compute dissimilarity between $x_i$ and other points $x_j$'s to obtain $D := \{d(x_i, x_j)\}_{j=1}^N$. Assuming there is no ties, i.e., $d(x_i, x_k) \neq d(x_i, x_j)$ for all $k \neq j$, then, the rank of $d(x_i, x_j)$ in the set $D$ can be computed as $r(d_{ij}) := \sum_k \mathbb{1}_{\{d_{ik} \leq d_{ij}\}}$. Moreover, the cumulative function $F(x_i, x_j)$ is the normalized rank of $d(x_i, x_j)$ in the set $D$. That is, $F(x_i, x_j) = r(d_{ij})/N = \frac{1}{N} \sum_k \mathbb{1}_{\{d_{ik} \leq d_{ij}\}}$.*

This remark allows us to intuitively understand the dissimilarity cumulative function through the lens of ranking. Hence, for a given reference point, we can know its "probabilistic distance" to the other points by using dissimilarity rankings to compute the cumulative function. This is illustrated in Fig. 3.

Using the cumulative function of Eq. (1), we define the perception coherence level as follows.

**Definition 3.2 (Perception coherence level)** *The perception coherence level of $f_2$ with respect to (w.r.t.) $f_1$ at $x \in \mathcal{X}$ is defined as*

$$\phi_{f_1, f_2}(x) := 1 - \mathbb{E}_X \left[ |F_1(x, X) - F_2(x, X)| \right]. \tag{2}$$

**Remark 3.3** *The coherence level is always valued in $[0, 1]$. Indeed, $0 \leq |F_1(x, x') - F_2(x, x')| \leq 1$ for all $x, x' \in \mathcal{X}$. Hence, $0 \leq \mathbb{E}_X \left[ |F_1(x, X) - F_2(x, X)| \right] \leq 1$.*

**Proposition 3.1** *If $f_2$ is absolutely perception coherent with $f_1$ at $x$, then $\phi_{f_1, f_2}(x) = 1$.*

**Proof 3.1** *By definition, we have that $F_1(x, X) = \mathbb{P}_{X'} \left( d_1(x, X') \leq d_1(x, X) \right)$ and $F_2(x, X) = \mathbb{P}_{X'} \left( d_2(x, X') \leq d_2(x, X) \right)$. As $f_2$ is absolutely perception coherent with $f_1$ at $x$, we obtain that $d_1(x, X') \leq d_1(x, X) \Leftrightarrow d_2(x, X') \leq d_2(x, X)$. Thus, $\mathbb{P}_{X'} \left( d_1(x, X') \leq d_1(x, X) \right) = \mathbb{P}_{X'} \left( d_2(x, X') \leq d_2(x, X) \right)$, which leads to the result.*

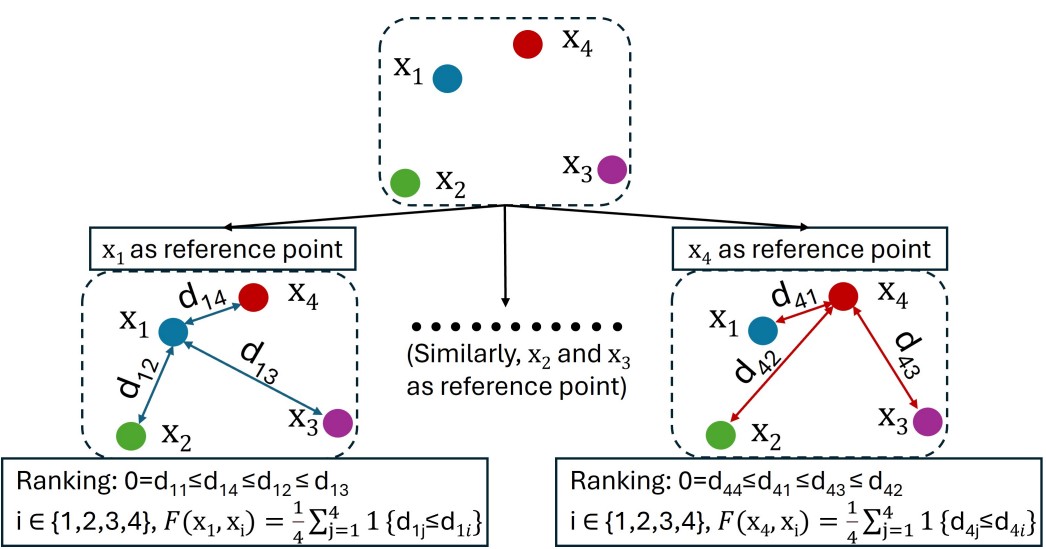

Figure 3: **Illustration of dissimilarity ranking (in feature space) in the case of 4 points.** Each of all the data points is considered as the reference point, and the dissimilarities to other data points are ranked to capture the cumulative distribution of dissimilarity (in feature space).

Proposition 3.1 tells that if we have an absolute perception coherence between the two models at a point $x$, the coherence level is equal to 1 at $x$. Now, a natural question to ask is whether the reverse statement holds. In short, our theoretical analysis yields a positive answer to this question. This will be presented in Section 4.

The coherence level naturally gives rise to the following definitions for the local and global perception coherence.

**Definition 3.3 (Local $\alpha$-perception coherence)** *We say that $f_2$ is $\alpha$-perception coherent with $f_1$ at $x \in \mathcal{X}$ if $\phi_{f_1,f_2}(x) \geq \alpha$.*

**Definition 3.4 (Global $\alpha$-perception coherence)** *We say that $f_2$ is globally $\alpha$-perception coherent with $f_1$ if $\mathbb{E}_X[\phi_{f_1,f_2}(X)] \geq \alpha$. That is, the global expected coherence level is at least $\alpha$.*

### 3.2 Practical implementation with a differentiable ranking operation

To implement our method, we use a simple Monte-Carlo sampling, processing by mini-batch, like any standard machine learning algorithm. Consider a training batch $\mathcal{B} := \{x_i\}_{i=1}^B$. For each $f \in \{f_1, f_2\}$ endowed with the metrics $d \in \{d_{f_1}, d_{f_2}\}$, we first compute the pair-wise dissimilarity $d(x_i, x_j)$ for all $i, j \in \{1, 2, \cdots, B\}$. Let $d_{ij}$ denote $d(x_i, x_j)$ for brevity. Let us denote $D_i^f(\mathcal{B}) := (d_{ij})_{j=1}^B$. Let $\mathcal{R}_i^f(\mathcal{B})$ be the ranking of $D_i^f(\mathcal{B})$. That is, the $j^{th}$ component of $\mathcal{R}_i^f(\mathcal{B})$ is the rank of $d_{ij}$ in $D_i^f(\mathcal{B})$. Notice that $\mathcal{R}_i^f(\mathcal{B})$ is a permutation of $\{1, 2 \cdots, B\}$ (so, $\mathcal{R}_i^f(\mathcal{B}) \in \mathbb{R}^B$). Based on Remark 3.2, intuitively, with sufficiently large $B$[1], for each $f \in \{f_1, f_2\}$, we have

$$\frac{\mathcal{R}_i^{f_1}(\mathcal{B})}{B} \approx (F_1(x_i, x_j))_{j=1}^B \ , \ \frac{\mathcal{R}_i^{f_2}(\mathcal{B})}{B} \approx (F_2(x_i, x_j))_{j=1}^B \ .$$

To maximize the perception coherence level, we need to somehow minimize the discrepancy between $F_1$ and $F_2$. Thus, empirically, we can minimize

$$\mathcal{L}(f_1, f_2; \mathcal{B}) := \frac{1}{B} \sum_{i=1}^B \left\| \frac{\mathcal{R}_i^{f_1}(\mathcal{B})}{B} - \frac{\mathcal{R}_i^{f_2}(\mathcal{B})}{B} \right\|^2 = \frac{1}{B^3} \sum_{i=1}^B \left\| \mathcal{R}_i^{f_1}(\mathcal{B}) - \mathcal{R}_i^{f_2}(\mathcal{B}) \right\|^2 \ .$$

---

[1]Through our qualitative observation, for $B$ from $\sim 10$, results start to stabilize. See Section 5.5 for results and discussions.

We use here the squared Euclidean norm for its simplicity, but other metrics can be used to compute the difference between $\mathcal{R}_i^{f_1}(\mathcal{B})$ and $\mathcal{R}_i^{f_2}(\mathcal{B})$. Besides, notice that each $i \in \{1, 2 \cdots, B\}$ corresponds to $x_i$ being the reference point. Hence, averaging over $i$ in the batch allows for maximizing the global expected coherence level (the expectation is taken over different reference points, see Definition 3.4 for recalling). However, the ranking operation is not differentiable. Hence, we propose a simple soft ranking operation based on sigmoid function. Assuming that there is no ties, i.e., $d_{ij} \neq d_{ik}$, for all $j \neq k$, that generally holds in practice. In this case, we notice that the rank of $d_{ij}$ in $D_i(\mathcal{B})$ can be computed as $r(d_{ij}) = \sum_{k=1}^{B} \mathbb{1}_{\{d_{ik} \leq d_{ij}\}}$. Hence, we can soften the ranking operation as follows

$$\widetilde{r}(d_{ij}) = \sum_{k=1}^{B} \Lambda \left( \frac{d_{ij} - d_{ik}}{\tau} \right) \ , \tag{3}$$

where $\Lambda$ is the sigmoid function and $\tau < 1$ is the temperature playing the role of scaling. The above function approaches the step function as $\tau$ tends to 0. Hence, this allows us to conveniently approximate the ranking operation. Let $\widetilde{\mathcal{R}}_i(\mathcal{B})$ be the corresponding ranking obtained with this softened version. So, our loss function can be written as,

$$\mathcal{L}_{\text{ours}}(f_1, f_2; \mathcal{B}) := \frac{1}{B^3} \sum_{i=1}^{B} \left\| \widetilde{\mathcal{R}}_i^{f_1}(\mathcal{B}) - \widetilde{\mathcal{R}}_i^{f_2}(\mathcal{B}) \right\|^2 \ . \tag{4}$$

## 4 Theoretical aspects

In this section, we first investigate in Section 4.1 the convergence behavior of the estimator of perception coherence level using mini-batches. Then, we derive theoretical insights about the local perception coherence at a given point $x$ in Section 4.2. Next, we also investigate about the global coherence in Section 4.3. Finally, we study the stability of the perception coherence around a given local region in Section 4.4. Throughout our theoretical analysis, we make the following assumptions.

**Assumption 4.1 (Measurability of Distribution Function)** *For $i = 1, 2$, the dissimilarity function $d_i : \mathcal{X} \times \mathcal{X} \to \mathbb{R}$ is measurable, and the marginal distribution $\mathcal{D}$ is such that the function $F_i(x, x') := \mathbb{P}_{X \sim \mathcal{D}}\big(d_i(x, X) \leq d_i(x, x')\big)$ is jointly measurable in $(x, x') \in \mathcal{X} \times \mathcal{X}$.*

**Assumption 4.2** *Let $X$ and $X'$ be the i.i.d random variables following the law $\mathcal{D}_{\mathcal{X}}$. We assume that for any $x \in \mathcal{X}$, $\mathbb{P}_{X, X'}\left(|F_2(x, X') - F_1(x, X')| = |F_2(x, X) - F_1(x, X)|\right) = 0$. We remark that this assumption is reasonable in the context of continuous variable modeling.*

### 4.1 Expected Convergence of the Mini-Batch Estimator

Recall from Eq. (2) that the *perception coherence level* of $f_2$ with respect to (w.r.t.) $f_1$ at $x \in \mathcal{X}$ is defined as

$$\phi_{f_1, f_2}(x) := 1 - \mathbb{E}_{X \sim \mathcal{D}}\Big[\big|F_1(x, X) - F_2(x, X)\big|\Big].$$

Averaging over the data distribution yields the *global perception coherence*,

$$\mathbb{E}_X[\phi_{f_1, f_2}(X)] = 1 - \mathbb{E}_{X_1, X_2 \sim \mathcal{D}}\Big[\big|F_1(X_1, X_2) - F_2(X_1, X_2)\big|\Big].$$

Maximizing the global perception coherence is equivalent to minimizing the following complementary quantity, which we call the **Difference Coefficient**:

$$\text{DC} := 1 - \mathbb{E}_X[\phi_{f_1, f_2}(X)] = \mathbb{E}_{X_1, X_2}\Big[\big|F_1(X_1, X_2) - F_2(X_1, X_2)\big|\Big].$$

Intuitively, DC measures the discrepancy between the rankings induced by $f_1$ and $f_2$ over the data distribution (i.e., discrepancy between $F_1$ and $F_2$).

**Mini-batch approximation.** In practice, computing DC exactly is infeasible, as it requires access to the full distribution (or the entire dataset). Instead, we approximate it using i.i.d. samples drawn in mini-batches. The corresponding empirical estimator is defined as follows.

**Definition 4.1 (Empirical Estimator)** *Let $\mathcal{B} = \{X_1, \ldots, X_B\}$ be a mini-batch of $B$ samples drawn i.i.d. from $\mathcal{D}$. For each $i \in \{1, 2\}$, define the empirical distribution function*

$$\widehat{F}_{i,B}(x, x') := \frac{1}{B} \sum_{k=1}^{B} \mathbb{1}\{d_i(x, X_k) \leq d_i(x, x')\}.$$

*Then the empirical difference coefficient is*

$$\widehat{\mathrm{DC}}_B := \frac{1}{B^2} \sum_{i=1}^{B} \sum_{j=1}^{B} \left| \widehat{F}_{1,B}(X_i, X_j) - \widehat{F}_{2,B}(X_i, X_j) \right|.$$

This estimator depends only on the samples in the mini-batch and provides a tractable approximation to DC. Before analyzing its convergence properties, we first verify that the inner terms are unbiased estimators of the population functions.

**Proposition 4.1 (Unbiasedness)** *For each fixed $i \in \{1, 2\}$ and any pair $(x, x') \in \mathcal{X}^2$, we have*

$$\mathbb{E}_{\mathcal{B}}\left[ \widehat{F}_{i,B}(x, x') \right] = F_i(x, x').$$

*In particular, $\widehat{F}_{i,B}$ is an unbiased estimator of $F_i$.*

**Proof 4.1** *By linearity of expectation,*

$$\mathbb{E}\left[ \widehat{F}_{i,B}(x, x') \right] = \frac{1}{B} \sum_{k=1}^{B} \mathbb{E}_{X_k \sim \mathcal{D}}[\mathbb{1}\{d_i(x, X_k) \leq d_i(x, x')\}] = \mathbb{E}_{X \sim \mathcal{D}}[\mathbb{1}\{d_i(x, X) \leq d_i(x, x')\}]$$

$$= \mathbb{P}_{X \sim \mathcal{D}}(d_i(x, X) \leq d_i(x, x')) = F_i(x, x').$$

**Convergence of the mini-batch estimator.** We now establish that $\widehat{\mathrm{DC}}_B$ converges to the true DC at the standard Monte Carlo rate.

**Theorem 4.1 (Expected Convergence Rate)** *Let $\widehat{\mathrm{DC}}_B$ be the empirical estimator of* DC *based on a batch of size $B$. Then there exists a constant $C > 0$, independent of $B$, such that*

$$\mathbb{E}_{\mathcal{B}}\left[ \left| \widehat{\mathrm{DC}}_B - \mathrm{DC} \right| \right] \leq \frac{C}{\sqrt{B}}.$$

*Consequently, $\widehat{\mathrm{DC}}_B \to \mathrm{DC}$ in expectation at the rate $O(1/\sqrt{B})$ as $B \to \infty$.*

**Proof 4.2** *The detailed proof is provided in Appendix A.*

**Discussion.** Theorem 4.1 ensures that the estimation error decays proportionally to $1/\sqrt{B}$. In other words, enlarging the mini-batch size systematically reduces the variance of the estimator, thereby improving the accuracy of the approximation to DC. This theoretical guarantee is strongly supported by our experiments, that will be shown in Section 5.5 (Fig. 7 and Table 5). More precisely, we shall show that the empirical estimation error exhibits the predicted scaling behavior with respect to $B$.

## 4.2 Local perception coherence

Theorems 4.2 and 4.3 provide insight into how well the student model $f_2$ mimics the teacher model $f_1$, given the local perception coherence level.

**Theorem 4.2 (Local rank preservation)** *Assume that $f_2$ is $\alpha$-perception coherent with $f_1$ at $x \in \mathcal{X}$ (i.e. $\phi_{f_1,f_2}(x) \geq \alpha$). Then, for any $\varepsilon_2 > \varepsilon_1 > 0$, we have*

$$\mathbb{P}_{X,X'}\left(|F_2(x,X') - F_2(x,X)| \geq \varepsilon_2 \;\Big|\; |F_1(x,X') - F_1(x,X)| \leq \varepsilon_1\right) \leq \frac{C_{\varepsilon_1}(1-\alpha)}{\varepsilon_2 - \varepsilon_1} \;, \tag{5}$$

*where $C_{\varepsilon_1}$ is a positive constant depending on $\varepsilon_1$.*

The proof is in Appendix B.1. Consider the case where $\alpha$ approaches 1, then the right-hand side (RHS) of Eq. (5) approaches 0. Theorem 5 shows that, if we pick up two random points $x_1$ and $x_2$ such that the teacher model $f_1$ perceives them to have close dissimilarities w.r.t. a given point $x$ (i.e., their dissimilarities differ by at most $\varepsilon_1$), then there is a high chance that it is also the case for the student model $f_2$. That is, $f_2$ also perceives that their dissimilarities differ by at most $\varepsilon_1$. This is because if $\alpha = 1$, the RHS is equal to 0 for any $\varepsilon_2 > \varepsilon_1$. While this theorem tells us only about the difference between the dissimilarities (absolute value), Theorem 4.3 also considers the ordering of the dissimilarities.

**Theorem 4.3 (Local relative order preservation)** *Assume that $f_2$ is $\alpha$-perception coherent with $f_1$ at $x \in \mathcal{X}$ ($\phi_{f_1,f_2}(x) \geq \alpha$). Then, for any $\varepsilon > 0$, we have*

$$\mathbb{P}_{X,X'}\left(F_2(x,X') - F_2(x,X) \geq \varepsilon \;\big|\; F_1(x,X') \leq F_1(x,X)\right) \leq C(1-\alpha)/\varepsilon \;, \tag{6}$$

*where $C$ is a positive constant.*

The proof of this theorem can be found in Appendix B.2. Notice that the RHS of Eq. (6) approaches 0 as $\alpha$ approaches 1 for any $\varepsilon > 0$. Hence, in this case, Theorem 4.3 shows that, if we pick up two random points $x_1$ and $x_2$ such that the teacher model $f_1$ perceives $x_1$ to be more similar to $x$ than $x_2$ (i.e., $F_1(x,x_1) \leq F_1(x,x_2)$), then probability that $f_2$ perceives $x_1$ to be less similar to $x$ than $x_2$ is upper bounded. In the case where $\alpha = 1$, this probability is 0, that can be formally stated as follows.

**Corollary 4.1** *Assume that $f_2$ is 1-perception coherent with $f_1$ at $x \in \mathcal{X}$ ($\phi_{f_1,f_2}(x) = 1$). Then, for any $\varepsilon > 0$, we have*

$$\mathbb{P}_{X,X'}\left(F_2(x,X') - F_2(x,X) \geq \varepsilon \;\big|\; F_1(x,X') \leq F_1(x,X)\right) = 0 \;.$$

Note that this holds for any $\varepsilon > 0$. Thus, by letting $\varepsilon \to 0$, the above corollary tells that, the probability that $f_2$ perceives $x_1$ to be less similar to $x$ than $x_2$ is null. That is, the student model $f_2$ learns to have the same perception as the teacher model $f_1$.

## 4.3 Global perception coherence

In the above section, we analyzed the perception coherence at a reference point $x$, knowing that the local coherence level at $x$ is at least $\alpha$. Now, a natural question concerning the global coherence follows: *What conclusions can be drawn from the global expected coherence level when the reference point $x$ is not fixed?* Theorem 4.4 formally answers this question.

**Theorem 4.4** *Assume that $f_2$ is globally $\alpha$-perception coherent with $f_1$ (i.e., $\mathbb{E}_X[\phi_{f_1,f_2}(X)] \geq \alpha$). Let $X_1$, $X_2$ and $X_3$ be i.i.d. random variables following the law $\mathcal{D}_\mathcal{X}$. Then,*

1. **Global rank preservation.** *For any $\varepsilon_2 > \varepsilon_1 > 0$, we have*

$$\mathbb{P}\left(|F_2(X_1,X_2) - F_2(X_1,X_3)| \geq \varepsilon_2 \;\Big|\; |F_1(X_1,X_2) - F_1(X_1,X_3)| \leq \varepsilon_1\right) \leq \frac{C_{\varepsilon_1}(1-\alpha)}{\varepsilon_2 - \varepsilon_1} \;,$$

   *where $C_{\varepsilon_1}$ is a positive constant depending only on $\varepsilon_1$.*

2. **Global relative order preservation.** *For any $\varepsilon > 0$, we have*

$$\mathbb{P}\left(F_2(X_1,X_2) - F_2(X_1,X_3) \geq \varepsilon \;\Big|\; F_1(X_1,X_2) \leq F_1(X_1,X_3)\right) \leq \frac{C(1-\alpha)}{\varepsilon} \;,$$

   *where $C$ is a positive constant.*

The proof of this theorem can be found in Appendix C.1. Theorem 4.4 shows that, even without fixing the reference point, knowing the global expected coherence guarantees a bound. This result can be interpreted similarly to the local perception case in Theorems 4.2 and 4.3.

### 4.4 Stability of perception coherence around a local region

We now study the fluctuation around a local region $\mathcal{A} \subseteq \mathcal{X}$. We seek to answer the following question: *If the local perception coherence level is at least $\alpha$ in the region $\mathcal{A}$, how stable is the coherence level under perturbations around $\mathcal{A}$?* In short, our theoretical result in Theorem 4.5 will show that the fluctuation of the coherence is bounded by the perturbation level around $\mathcal{A}$. For this, we make the following assumptions.

**Assumption 4.3** *We make the following assumptions.*

- *The dissimilarities of both models are symmetric and satisfy the triangular inequality.*

- *For $\delta \geq 0$, let $\rho(\delta) = \max_{i \in \{1,2\}} \sup_{x \in \mathcal{X}, d \in \mathbb{R}} \mathbb{P}_X (d_i(x, X) \in [d, d+\delta])$. We assume that $\rho(\delta) = O(\delta)$ as $\delta \to 0$.*

- *For $i \in \{1, 2\}$, $x, x' \in \mathcal{X}$, $d_i(x, x') = O(F_i(x, x'))$ as $F_i(x, x') \to 0$.*

Let $F(x', x) = \max(F_1(x', x), F_2(x', x))$. Intuitively, a small value of $F(x', x)$ signifies that $x$ and $x'$ are probabilistically close to each other in the feature spaces of both models $f_1$ and $f_2$. Then, the relation between the fluctuation of the perception coherence and the perturbation around $\mathcal{A}$ is stated as follows.

**Theorem 4.5** *Consider a non-empty set $\mathcal{A} \subseteq \mathcal{X}$ (with $\mathbb{P}_X (X \in \mathcal{A}) > 0$) such that $\phi_{f_1, f_2}(x) \geq \alpha$, $\forall x \in \mathcal{A}$. For $0 < \delta \leq 1$, let $\mathcal{A}_\delta = \{x \in \mathcal{X} : \min_{x' \in \mathcal{A}} F(x', x) \leq \delta\}$. Then, for any $0 < \varepsilon \leq \alpha$, we have*

$$\mathbb{P}_X \left( \phi_{f_1, f_2}(X) \leq \alpha - \varepsilon \ \Big| \ X \in \mathcal{A}_\delta \right) \leq \frac{O(\delta)}{\varepsilon C_\mathcal{A}} \,, \tag{7}$$

*where $C_\mathcal{A}$ is a positive constant depending on $\mathcal{A}$.*

The proof can be found in Appendix D. Notice that $\mathcal{A}_\delta$ is the perturbed set around $\mathcal{A}$ at *perturbation level $\delta$ ($\mathcal{A} \subseteq \mathcal{A}_\delta$)*. Theorem 4.5 states that if the coherence level in $\mathcal{A}$ is at least $\alpha$, then the probability that the coherence level of a point around $\mathcal{A}$ drops below $\alpha - \varepsilon$ is bounded above by $O(\delta)$. This holds for any $\varepsilon > 0$, indicating that the coherence level does not decrease significantly as we expand the region $\mathcal{A}_\delta$ (by increasing $\delta$). Thus, as the perturbation $\delta \to 0$, the upper bound also tends to zero, characterized by $O(\delta)$. This ensures the stability of coherence level within each local region.

## 5 Experiments

In this section, we first conduct experiments on $2D$ and $3D$ data for proof-of-concept (Section 5.1). Next, in Section 5.2, we present a stylized empirical study showing the correlation between the perception coherence level during transfer and the performance of downstream classification task. Then, in Section 5.3, we conduct experiments on lightweight settings to evaluate the quality of the learned representation through retrieval tasks. Besides, we demonstrate how our method can boost the performance of student models in classification task in Section 5.4. Note that in all experiments conducted in Sections 5.3 and 5.4, the student models are smaller than the teacher models[2], to meet practical requirements. Finally, we shall also conduct ablation study in Sections 5.5 and 5.6. Without further mention, we use cosine dissimilarity as the dissimilarity metric on neural networks (defined in Appendix E). For brevity, the details for all experiments can also be found in Appendix E.

---

[2]See Appendix F for detailed computational cost comparison of different pairs of teacher-student models.

### 5.1 Proof-of-concept with experiments on 2D and 3D toy examples

In this section, we conduct experiments on $2D$ and $3D$ data, without neural networks, to qualitatively demonstrate how our method works. For this, we use a reference data configuration $\{x_i^1\}_{i=1}^N$ (can be seen as the teacher configuration, $N$ is the number of data points). Then, we randomly initialize another configuration $\{x_i^2\}_{i=1}^N$ (student configuration), where each point $x_i^2$ is associated with a fixed point $x_i^1$. More formally, for each $i \in \{1, 2 \cdots N\}$, $f_1(i) = x_i^1$ (fixed) and $f_2(i) = x_i^2$. We then apply our loss function from Eq. (4), where the dissimilarity metric is the Euclidean distance. All experiment details are in Appendix E.1. The evolution of the student configurations in different setups are shown in Fig. 4 and 5. We intentionally set the teacher and student configurations in different scales of magnitude to show the effectiveness of the method (teacher configuration has data points in scale $\sim 1$ whereas student configuration is in scale $\sim 10$). We see that our method results in a very smooth configuration transfer, despite the scale difference. From Fig. 4, we remark that, while the exact geometry is not completely preserved, the learned configuration preserves a global structural coherence. The same phenomenon is also observed in the case where teacher and student configurations have different dimensions ($3D \rightarrow 2D$) in Fig. 5. Another experiment with more clusters is in Appendix E.1. Thus, this qualitatively proves the effectiveness of our method.

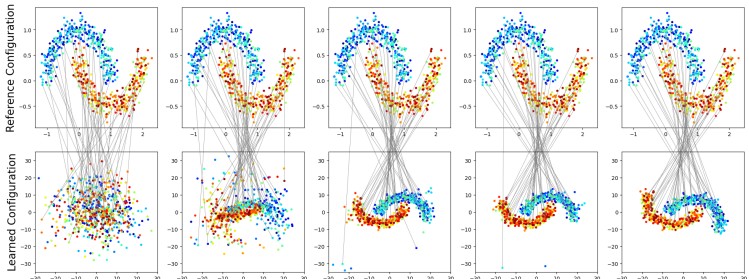

Figure 4: Transferring process on $2D$ datasets. **Top row** represents the teacher configuration (fixed), **bottom row** represents the evolution of the student configuration along the transfer process (from left to right). Each gray line associates $x_i^2$ with $x_i^1$ (for the same $i$). We observe that the learned configuration preserves a global structural coherence without preserving completely the geometry.

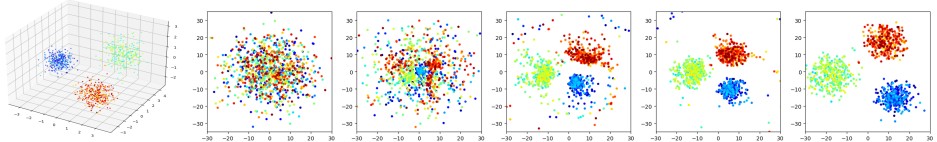

(a) $3D$ reference configuration. (b) Evolution along the transferring process of learned configuration in $2D$.

Figure 5: Transferring process from $3D$ to $2D$ data.

### 5.2 A Stylized Empirical Study of Perception Coherence and Downstream Classification

To show the correlation between perception coherence and downstream task performance, we study its relationship with classification performance in a controlled setting. We consider a two-moon binary classification dataset with 800 samples, randomly split into 400 training and 400 test points. For **teacher model**[3], a neural network with two hidden layers followed by a softmax classifier is trained in a fully supervised manner, using the labeled training set. After convergence, the teacher achieves perfect accuracy on both training and test sets.

**Student model and unsupervised transfer.** For simplicity, the student network shares the same architecture as the teacher model. Using our proposed unsupervised transfer method, we distill the representation from the teacher's penultimate layer into the student, relying only on unlabeled training data. Importantly, the final softmax layer of the student is excluded from this transfer process. During training, we save student checkpoints at ten different epochs, denoted by $\{S_k\}_{k=1}^{10}$.

---

[3]For brevity, please refer to Appendix E.2 for model details, including training details.

**Perception coherence.** For each checkpoint $S_k$, we compute the perception coherence coefficient on the training set, yielding values $\{\text{PC}_k\}_{k=1}^{10}$.

**Downstream classification.** To evaluate the usefulness of the transferred representations, we freeze all feature layers of each student model in $\{S_k\}_{k=1}^{10}$ and train only the final softmax layer in a supervised fashion on the labeled training set. This protocol isolates the quality of the learned representation: higher accuracy directly reflects task-relevant features. After training, by testing the trained models on the test set, we obtain the test accuracies $\{\text{Acc}_k\}_{k=1}^{10}$ corresponding to different checkpoints $\{S_k\}_{k=1}^{10}$.

Table 1 and Fig. 6 show the relation between the perception coherence values and corresponding downstream classification accuracies (i.e., $\{\text{PC}_k\}_{k=1}^{10}$ *v.s.* $\{\text{Acc}_k\}_{k=1}^{10}$). We observe a strong positive relationship between perception coherence and downstream performance: student models with higher PC consistently achieve better classification accuracy. Quantitatively, the Pearson correlation coefficient between training PC and test accuracy is 0.920. These results establish a direct task-level connection between perception coherence and classification error. Even in this stylized setting, improved coherence of the learned representation leads to higher linear separability and better downstream accuracy. This supports the interpretation of perception coherence as a meaningful proxy for representation transferring quality (without the need for label).

Table 1: Training perception coherence *versus* downstream test accuracy for student models at different training epochs.

| Training perception coherence level | 0.841 | 0.877 | 0.889 | 0.901 | 0.911 | 0.921 | 0.931 | 0.940 | 0.949 | 0.956 |
|---|---|---|---|---|---|---|---|---|---|---|
| Test accuracy (%) | 82.00 | 83.50 | 83.25 | 85.75 | 86.75 | 88.00 | 86.75 | 89.25 | 87.00 | 90.00 |

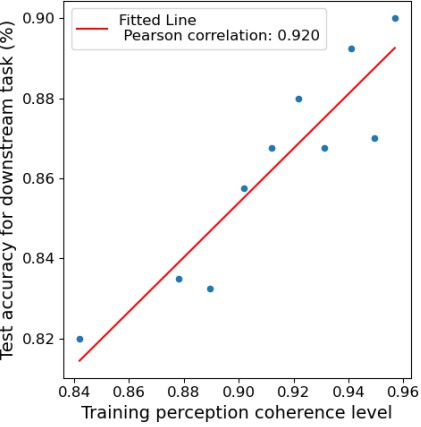

Figure 6: Training perception coherence *versus* downstream test accuracy for student models at different training epochs.

## 5.3 Learning Metric Evaluation on Neural Networks

In this section, we perform experiments on CIFAR10 (Krizhevsky et al., 2009) and the subset of the first 30 classes of CUB-200 (Wah et al., 2011) with very small student models. Following the previous works (e.g. (Passalis & Tefas, 2018a; Passalis et al., 2020b;a)), we evaluate the quality of the learned representation using the content-based image retrieval setup. We used the official code released with the work of Passalis et al. (2020a) to evaluate different metrics, including the interpolated mean Average Precision (mAP) (at the standard 11-recall points) and the top-$k$ precision (Christopher et al., 2008). For a fair comparison, we use the same teacher and student model architecture as in Passalis et al. (2020a). The student models are very small, composed of only 3 convolutional layers and a fully connected layer. All details are in Appendix E.3. To conduct the experiments, for each dataset CIFAR10 and CUB-200, we first train the teacher model on the training set. Then, the transfer set is only the training set without label. Then, the retrieval setup is used to evaluate the quality of the learned representation. The database contains the training set and the test set is

used for evaluating the performance of each model. If the student is trained to learn useful features, then the test examples should be close to the training examples (database) of the same class. Hence, a higher retrieval score indicates a better quality of the learned representation. The results for CIFAR10 and CUB-200 are shown in Tables 2 and 3, respectively. Our method is also compared with classical methods, including the standard knowledge distilling (KD) (Hinton et al., 2015), FitNet (Romero et al., 2014), metric knowledge transfer (MKT) (Yu et al., 2019), probabilistic knowledge transfer (PKT) (Passalis & Tefas, 2018a) and Heterogeneous Knowledge Distillation (HKD) (Passalis et al., 2020a). The results displayed for all these methods in Tables 2 and 3 are extracted from Passalis et al. (2020a). Therein, one used the same teacher and student models. For FitNet, MKT, PKT, there are two versions: the first one applies only on a single penultimate layer, while the second one applies on multiple layers (denoted by -H).

Table 2: Learning Metric Evaluation on CIFAR10.

| Multi-layer or single layer distilling | Method | Auxiliary Model | mAP | top-100 |
|---|---|---|---|---|
| Teacher Model (ResNet-18) | | | 91.94 | 93.50 |
| Single Layer Distilling | **Ours** | No | **54.25** | **65.00** |
| | KD (Hinton et al., 2015) | No | 40.53 | 58.56 |
| | FitNet (Romero et al., 2014) | No | 48.99 | 62.42 |
| | MKT (Yu et al., 2019) | No | 38.20 | 52.72 |
| | PKT (Passalis & Tefas, 2018a) | No | 51.56 | 62.50 |
| Multi-layer Distilling | FitNet-H | No | 46.46 | 60.59 |
| | MKT-H | No | 43.99 | 57.63 |
| | PKT-H | No | 51.73 | 63.01 |
| | HKD (Passalis et al., 2020a) | Yes | 53.06 | 64.24 |

Table 3: Learning Metric Evaluation on a subset of the first 30 classes of CUB200.

| Multi-layer or single layer distilling | Method | Auxiliary Model | mAP | top-10 |
|---|---|---|---|---|
| Teacher Model (ResNet-18) | | | 67.48 | 74.38 |
| Single Layer Distilling | **Ours** | No | **28.42** | **36.55** |
| | KD (Hinton et al., 2015) | No | 18.55 | 26.57 |
| | FitNet (Romero et al., 2014) | No | 15.98 | 23.41 |
| | MKT (Yu et al., 2019) | No | 13.39 | 20.59 |
| | PKT (Passalis & Tefas, 2018a) | No | 18.57 | 26.70 |
| Multi-layer Distilling | FitNet-H | No | 15.37 | 22.61 |
| | MKT-H | No | 15.39 | 22.76 |
| | PKT-H | No | 17.77 | 25.39 |
| | HKD (Passalis et al., 2020a) | Yes | 19.01 | 27.67 |

From Tables 2 and 3, we observe that MKT method provides the worst results. This method is composed of 2 main ingredients: (1) minimizing the distance between the student and teacher features and (2) enforcing the student model to result in similar distances between the data points as the teacher model (in the feature space). Hence, in our setting of very small student models, this brute force approach provides the worst results. Our method also outperforms by a large gap the standard approaches of KD (using the soft label) or FitNet (using the distance between the student and teacher features). More advanced methods such as PKT (based on a kernel method) performs better, but it is still outperformed by our method. Besides, HKD is a more advanced version of PKT, where one uses a particular mechanism to transfer the features. However, our method still provides better results. Notice that our method is applied only on the penultimate layer, but outperforms the other methods using multiple layers. Especially on CUB-200, our method outperforms the others by a wide margin. This proves the effectiveness of our method for feature representation transferring.

### 5.4 Our method as a guide for knowledge distilling: comparing with other methods of feature representation transfer in the context of classification

In the above section, we show how our method helps the student model to produce useful features. In this section, we demonstrate that by transferring the feature representation, our method boosts significantly the performance of the student model in the context of the classification task on CIFAR100 (Krizhevsky, 2009). While different methods have been proposed and tailored specifically for the classification, we focus mainly on state-of-the-art feature representation transferring methods. Notice that different methods (such as FitNet (Romero et al., 2014)) apply the transferring on different intermediate layers. However, we propose to apply our method only on the penultimate layer and on the logit one (before applying the softmax function). We also fix all the hyperparameters for training, without further tuning to show the simplicity of our method. The most recent VRM method (Zhang et al., 2025) introduces virtual relation matching and applies RandAug (Random Augmentation). For a fair comparison, we add RandAug to the inputs using the official code associated with this last work. Then, we apply our method under the same setting. All details can be found in Appendix E.4. While different methods have been proposed to effectively transfer knowledge between model of the same architecture type (e.g. ResNet50 → ResNet18), we intentionally choose pairs of models having different architecture types. More precisely, we use 3 pairs of teacher and student models: ResNet-50 → MobileNetV2 and ResNet-32x4 → ShuffleNetV1 / ShuffleNetV2. We follow the same setup and use the same pretrained teacher models officially released with the work of Tian et al. (2020) for fair comparisons.

Table 4: Test classification accuracy (%) on CIFAR100 using different knowledge distillation methods. Results of ReviewKD (Chen et al., 2021), TTM (Zheng & YANG, 2024) and VRM (Zhang et al., 2025) are extracted from the corresponding original papers, other methods are extracted from Huang et al. (2022), where one used the same setups.

| Method | ResNet-50 → MobileNetV2 | ResNet-32x4 → ShuffleNetV1 | ResNet-32x4 → ShuffleNetV2 |
|---|---|---|---|
| Teacher | 79.34 | 79.42 | 79.42 |
| Student | 64.6 | 70.5 | 71.82 |
| KD (Hinton et al., 2015) | 67.35±0.32 | 74.07±0.19 | 74.45±0.27 |
| FitNet (Romero et al., 2014) | 63.16±0.47 | 73.59±0.15 | 73.54±0.22 |
| VID (Ahn et al., 2019) | 67.57±0.28 | 73.38±0.09 | 73.40±0.17 |
| RKD (Park et al., 2019) | 64.43±0.42 | 72.28±0.39 | 73.21±0.28 |
| PKT (Passalis & Tefas, 2018a) | 66.52±0.33 | 74.10±0.25 | 74.69±0.34 |
| CRD (Tian et al., 2020) | 69.11±0.36 | 75.11±0.32 | 75.65±0.10 |
| ReviewKD (Chen et al., 2021) | 69.89 | 77.45 | 77.78 |
| DIST (Huang et al., 2022) | 68.66±0.23 | 76.34±0.18 | 77.35±0.25 |
| TTM (Zheng & YANG, 2024) | 69.59 | 74.37 | 76.55 |
| VRM (with RandAug) (Zhang et al., 2025) | **72.30** | 78.28 | **79.34** |
| Ours (with RandAug) | 71.10±0.10 | **78.59**±0.13 | 79.14±0.10 |

From Table 4, we see that our method boosts significantly the performance of the student models (when not using any KD technique). It also provides much better results than standard techniques such as KD or FitNet. This proves its effectiveness for extracting knowledge from the teacher model. Besides, our method systematically outperforms the PKT method based on kernel techniques (see Section 2). This reinforces our intuition that our method is more effective than applying the same kernel for all points, as discussed in Section 2. Other methods based on mutual information such as Variational Information Distillation (VID) is also outperformed by ours. The method of Contrastive Representation Distillation (CRD) is also based on an information-theoretic framework. Our method outperforms this last one. Moreover, it outperforms all the other methods on ResNet-32x4/ShuffleNetV1 benchmark. We also observe that our method achieves on-par performance compared to VRM, even though VRM includes additional components such as inter-class relation modeling and edge pruning strategies. Our approach intentionally uses a minimal setting, yet still matches VRM performance. All these remarks prove the effectiveness of our method for KD.

Finally, notice that some different knowledge distilling methods are tailored specifically for classification (e.g. Huang et al. (2022)). However, our main objective is to study methods for feature representation transferring, as this class of methods is more generic and applicable on a wider range of tasks (e.g. retrieval). Through our experiments, we aim to show that a good feature representation transfer allows us to effectively distill the knowledge from a neural network.

## 5.5 Ablation study: effect of mini-batch size

In Section 4.1, we established a theoretical result showing that the global perception coherence level (GPCL), when estimated using mini-batches of size $B$, converges in expectation to its true value at a rate of $O(1/\sqrt{B})$ (Theorem 4.1). We now turn to an empirical validation of this result. To this end, we employ the estimator defined in Definition 4.1 with varying batch sizes, and evaluate it on trained student models from the CIFAR10 and CUB200 experiments. The outcomes are summarized in Figure 7 and Table 5.

From Table 5, we observe that for very small batch sizes (e.g., $B = 4$), the estimated GPCL deviates significantly from the reference value, reflecting the limited information available in such small batches. As the batch size increases (e.g., $B = 16$ or $B = 64$), the estimated GPCL rapidly converges toward the true value. In fact, with $B = 32$ or $B = 64$, the estimator already provides a stable and reliable approximation. For instance, in the CIFAR10 case with 10,000 test examples, using $B = 64$ yields an accurate estimate with low variance, thereby confirming the predicted convergence rate with respect to batch size.

Table 5: Estimation of global perception coherence level, using different mini-batch sizes.

| Mini-batch size | 4 | 8 | 16 | 32 | 64 | 128 | 256 | full dataset |
|---|---|---|---|---|---|---|---|---|
| CIFAR10 | $0.8943 \pm 0.00098$ | $0.8377 \pm 0.00061$ | $0.8131 \pm 0.00032$ | $0.8021 \pm 0.00029$ | $0.7969 \pm 0.00028$ | $0.7944 \pm 0.00017$ | $0.7931 \pm 0.00014$ | **0.7919** |
| CUB200 subset | $0.8919 \pm 0.00271$ | $0.8312 \pm 0.00114$ | $0.8073 \pm 0.00085$ | $0.7959 \pm 0.00136$ | $0.7908 \pm 0.00081$ | $0.7878 \pm 0.00059$ | $0.7874 \pm 0.00057$ | **0.7859** |

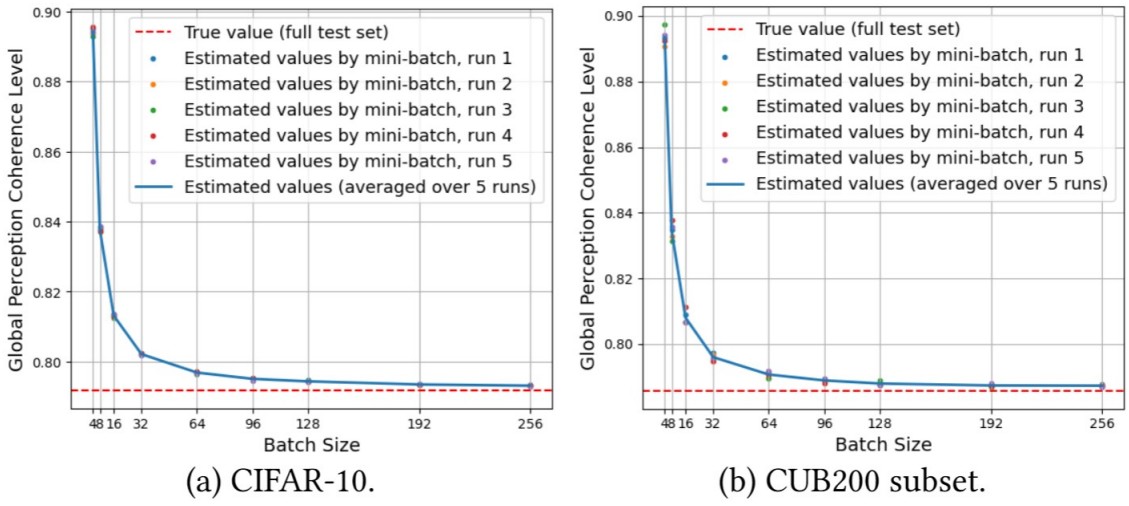

(a) CIFAR-10.  (b) CUB200 subset.

Figure 7: Estimated global perception coherence level at different batch sizes.

To further investigate the qualitative impact of batch size, we conduct experiments on toy datasets where representations are transferred from $\mathbb{R}^3 \to \mathbb{R}^2$ (Figure 8) and from $\mathbb{R}^2 \to \mathbb{R}^2$ (Figure 9). Here, we explicitly vary $B$ and analyze the resulting student configurations. When $B$ is too small (e.g., $B = 3$), the learned configuration fails to preserve the structural coherence of the reference configuration. This is expected, as such small batches provide only very limited relational information between points. Starting from $B = 7$, however, the learned configurations begin to recover the global structure, and from $B = 15$ onward, the results become stable.

These findings suggest that excessively large batch sizes are not necessary for capturing global coherence. Indeed, during training, each epoch involves drawing random mini-batches of size $B$, and within each batch, every reference point is compared to $B - 1$ others. Due to random sampling, across successive epochs each point is paired with different comparison sets. As a result, the model gradually aggregates sufficient global structural information, even when $B$ is relatively small compared to the dataset size. This explains why reliable approximations of GPCL and structurally coherent student configurations can be obtained without resorting to very large batch sizes.

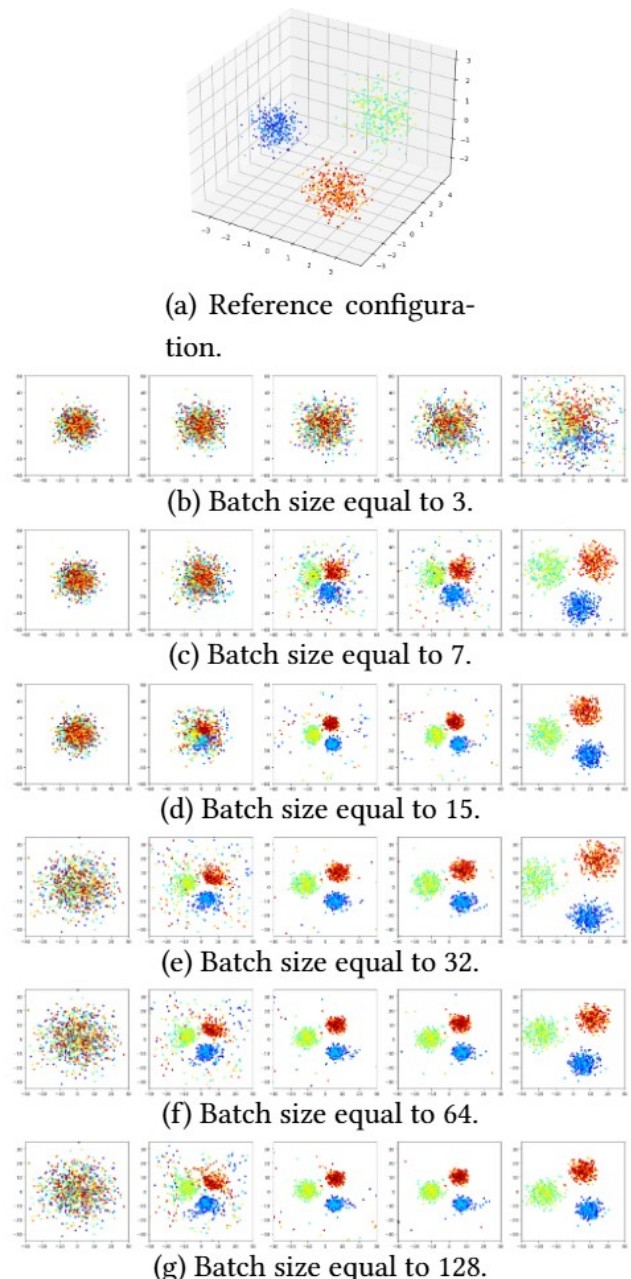

Figure 8: Ablation study of the effect of batch size on $3D \to 2D$ Gaussian dataset, where we train with different batch sizes. Plots from left to right in each row represent evolution of the learned configuration during training.

## 5.6 Ablation study: Student model size

In this section, we investigate the influence of student model size on transfer performance. Recall that in the CIFAR10 and CUB200 experiments, the baseline student architecture consists of three convolutional layers (**conv1/conv2/conv3**), each followed by a ReLU non-linearity, and a final fully connected layer. To explicitly examine the effect of model size, we construct progressively smaller student variants by truncating the network at different depths and applying Global Average Pooling (GAP)[4] over the two spatial dimensions to obtain the final representation:

---

[4]Formally, GAP: $\mathbb{R}^{H \times W \times C} \mapsto \mathbb{R}^C$, such that for $z \in \mathbb{R}^{H \times W \times C}$, for $k \in \{1, 2, \cdots, C\}$, $\text{GAP}(z)_k = \frac{1}{HW} \sum_{i=1}^{H} \sum_{j=1}^{W} z_{ijk}$.

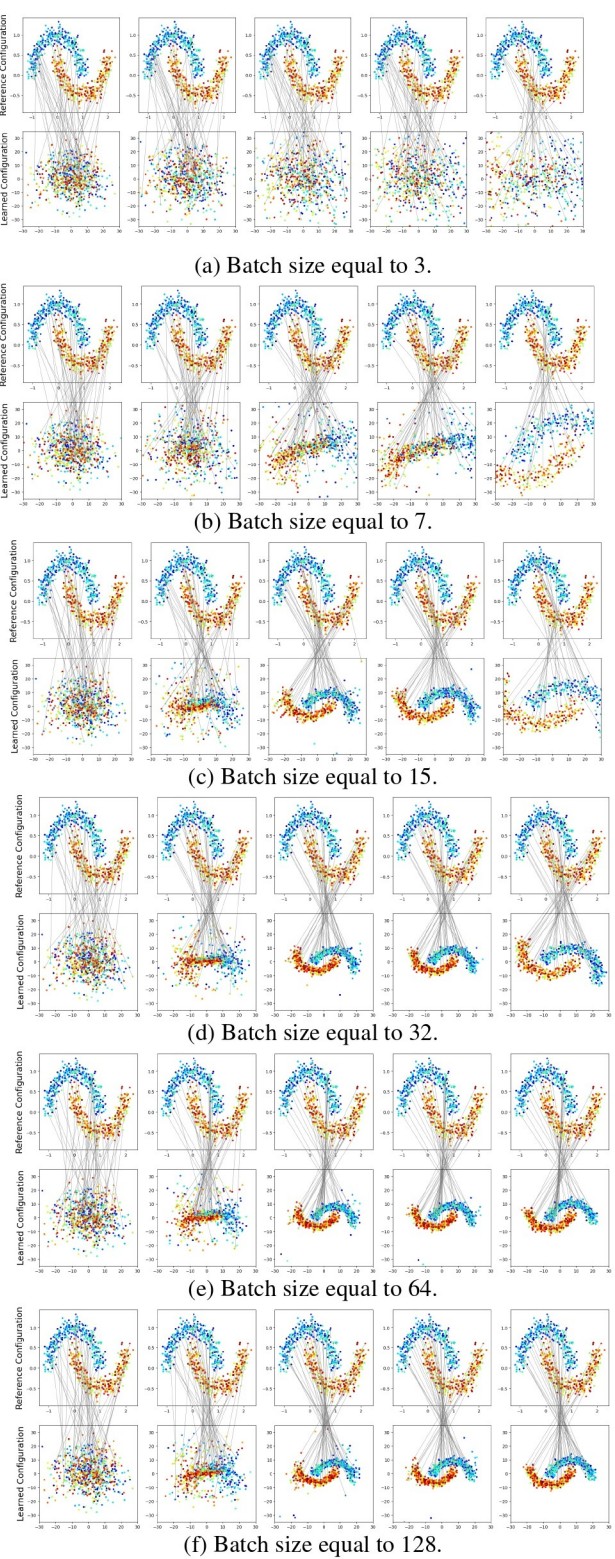

(a) Batch size equal to 3.

(b) Batch size equal to 7.

(c) Batch size equal to 15.

(d) Batch size equal to 32.

(e) Batch size equal to 64.

(f) Batch size equal to 128.

Figure 9: Ablation study of the effect of batch size on two-moon dataset, where we train with different batch sizes.

- **Model 1** (one layer): **conv1** + GAP.

- **Model 2** (two layers): **conv1** + **conv2** + GAP.

- **Model 3** (three layers): **conv1** + **conv2** + **conv3** + GAP.

- **Model 4**: the full student model (original architecture).

The results are reported in Table 6. For both datasets, we observe a consistent trend: as the student model becomes larger, the global perception coherence level (GPCL) increases. This indicates that deeper models are more capable of preserving the ranking structure induced by the teacher's feature space.

Moreover, we note a strong correlation between GPCL and downstream performance metrics such as mean Average Precision (mAP) and top-$k$ accuracy. In particular, models with higher GPCL systematically achieve better classification performance. This empirical evidence reinforces the role of GPCL as a meaningful and effective objective for guiding teacher–student learning. It also highlights a practical limitation: when the student is excessively small, its limited representational capacity hampers the preservation of structural coherence, leading to degraded performance.

Table 6: Performance of different models, with different sizes, with the same teacher model.

| Data set | Metric | One layer | Two layers | Three Layers | Full model |
|---|---|---|---|---|---|
| | Global PC Level (on test set) | 0.7015 | 0.7491 | 0.7700 | 0.7919 |
| CIFAR10 | mAP | 17.08 | 26.52 | 37.69 | 54.25 |
| | top-100 | 20.43 | 35.36 | 50.29 | 65.00 |
| | Global PC Level (on test set) | 0.7015 | 0.7266 | 0.7602 | 0.7859 |
| CUB200 subset | mAP | 6.63 | 8.66 | 18.88 | 28.42 |
| | top-10 | 10.35 | 13.38 | 27.28 | 36.55 |

## 6 Scope and Practical Considerations

**Scope.** Our method is intentionally designed for *heterogeneous* teacher–student settings, where the teacher and student may differ substantially in architecture and capacity. This scenario reflects practical deployment needs, in which knowledge is often transferred from a large, high-performance model to a smaller and structurally different model for efficiency or hardware constraints. In contrast, we are convinced that homogeneous settings (e.g., similar architectures) are typically well served by standard knowledge distillation (KD) approaches. Our goal is therefore not to replace or improve conventional KD in homogeneous cases, but rather to focus on practically relevant heterogeneous regime.

Moreover, our focus is to introduce a new measure, termed *perception coherence*, for feature representation transfer between models, ideally to enhance representation alignment specifically within the KD setting. Accordingly, our empirical comparisons are restricted to distillation-based baselines. Broader general-purpose representation learning frameworks, such as self-supervised representation learning, are beyond the scope of this study. Nevertheless, these methods are conceptually complementary to our framework, and their integration with the proposed approach represents a promising avenue for future investigation.

**Computational Complexity.** The theoretical per-batch complexity of the proposed loss is $\mathcal{O}(B^3)$ due to the computation and comparison of pairwise dissimilarities under soft ranking. However, the implementation relies entirely on batched tensor operations (distance matrix computation, sigmoid-based soft ranking, and squared loss), which are highly optimized on modern GPUs. Moreover, soft-ranking computations for different reference (anchor) points are independent and can be parallelized. With $N$ GPUs, the batch can be partitioned along the reference dimension, allowing each device to process approximately $B/N$ anchors. The final loss is aggregated via standard reduction operations (e.g., all-reduce), enabling near-linear scaling in practice and substantially reducing wall-clock training time.

Besides, it is worth emphasizing that, in many practical and industrial scenarios, lightweight models are trained on servers or high-performance hardware and subsequently deployed to edge devices for inference. Therefore, the training cost does not impact the deployment of the final student model on the edge devices.

**Practical Guidance for the Temperature Parameter $\tau$.** The temperature parameter $\tau$ in the soft-ranking formulation controls the smoothness of relative distance comparisons. Empirically, we find that values in the range $\{0.1, 0.2, 0.3\}$ work well for both teacher and student models. From a training-dynamics perspective, using a slightly larger $\tau$ for the student can facilitate optimization, as excessively small $\tau$ values may weaken gradient signals. Conversely, a smaller $\tau$ for the teacher encourages sharper relative ranking, since large $\tau$ values render the sigmoid function more linear and reduce ranking meaning[5]. In our experiments, we intentionally use the same $\tau$ across different teacher–student pairs for simplicity and fairness. Nevertheless, task-specific tuning of $\tau$ may further improve performance in practical applications.

## 7 Discussion and Conclusion

Preserving dissimilarity ranks constitutes a flexible requirement, as it does not enforce a complete replication of the teacher's geometry. Nevertheless, an important theoretical question remains open: *to what extent can a given student model preserve these rankings as a function of its representational capacity?* Addressing this question lies beyond the scope of the present work but represents a promising direction for future theoretical investigations.

**Why rankings?** Generally, in the teacher model's feature space, smaller distances (or dissimilarities) often reflect higher semantic similarity. By teaching the student model to preserve the relative rankings, it learns to order semantic dissimilarities in a way that aligns with how the teacher model perceives the input semantic dissimilarity. In order to replicate this ordering, the student model has to understand features that capture similar semantic content—thus encouraging the learning of meaningful and transferable representations.

By considering dissimilarity rankings, our method implicitly captures structural information about the underlying data manifold. This perspective can be viewed as a form of topology-aware representation transfer. Topology is concerned with invariant properties of a configuration under continuous deformations. Analogously, in our framework, as long as the relative distance orderings are preserved, the learned representation remains invariant under local or global distortions. This provides additional flexibility and stability, distinguishing our approach from geometry-preserving methods. A notable difference from classical topological approaches lies in the probabilistic framework that we introduce. While topology typically characterizes data configurations deterministically, our method relaxes this view by embedding ranking preservation in a probabilistic setting. Furthermore, by integrating neural networks into this framework, we open the path toward topology-aware knowledge transfer in deep learning.

In summary, this paper introduced an original method for representation transfer between models with arbitrary feature dimensions, based on the novel concept of *perception coherence*. The probabilistic formulation provides new theoretical insights into the knowledge distillation process, showing that in the ideal case the student perceives the input space in the same way as the teacher. Experiments confirm that the method enables efficient lightweight transfer while maintaining competitive performance. Importantly, the proposed framework is generic and can be applied with different dissimilarity metrics for the teacher and student models. This flexibility suggests promising applications in task-specific contexts, such as multi-domain or multi-modal transfer. We believe that this work paves the way for future advancements in knowledge distillation, especially those grounded in topological perspectives for deep learning.

---

[5]See Appendix E.5 for a brief analysis about the temperature.

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

## Appendix A    Convergence of the global perception coherence level estimated by mini-batch

We provide here a full proof of Theorem 4.1.

**Proof A.1** *Step 1: Decompose error*

*We note that*

$$\widehat{\mathrm{DC}}_B - \mathrm{DC} = \frac{1}{B^2} \sum_{i,j} \left( \left| \widehat{F}_{1,B}(X_i, X_j) - \widehat{F}_{2,B}(X_i, X_j) \right| - |F_1(X_i, X_j) - F_2(X_i, X_j)| \right)$$
$$+ \frac{1}{B^2} \sum_{i,j} |F_1(X_i, X_j) - F_2(X_i, X_j)| - \mathrm{DC} .$$

*Thus,*

$$\left| \widehat{\mathrm{DC}}_B - \mathrm{DC} \right| \leq \left| \frac{1}{B^2} \sum_{i,j} \left( \left| \widehat{F}_{1,B}(X_i, X_j) - \widehat{F}_{2,B}(X_i, X_j) \right| - |F_1(X_i, X_j) - F_2(X_i, X_j)| \right) \right|$$
$$+ \left| \frac{1}{B^2} \sum_{i,j} |F_1(X_i, X_j) - F_2(X_i, X_j)| - \mathrm{DC} \right| .$$

*We now take expectation on both sides:*

$$\mathbb{E} \left| \widehat{\mathrm{DC}}_B - \mathrm{DC} \right| \leq E_1 + E_2,$$

*where*

$$E_1 := \mathbb{E} \left[ \left| \frac{1}{B^2} \sum_{i,j} \left( \left| \widehat{F}_{1,B}(X_i, X_j) - \widehat{F}_{2,B}(X_i, X_j) \right| - |F_1(X_i, X_j) - F_2(X_i, X_j)| \right) \right| \right] ,$$
$$E_2 := \mathbb{E} \left[ \left| \frac{1}{B^2} \sum_{i,j} |F_1(X_i, X_j) - F_2(X_i, X_j)| - \mathrm{DC} \right| \right] .$$

*Step 2: Bound $E_1$*

*We use the reverse triangle inequality: for any real numbers $a, b$,*

$$||a| - |b|| \leq |a - b|.$$

*Applying this, we obtain:*

$$E_1 = \mathbb{E}\left[\left|\frac{1}{B^2}\sum_{i,j=1}^{B}\left(\left|\widehat{F}_{1,B}(X_i,X_j) - \widehat{F}_{2,B}(X_i,X_j)\right| - |F_1(X_i,X_j) - F_2(X_i,X_j)|\right)\right|\right]$$

$$\leq \mathbb{E}\left[\frac{1}{B^2}\sum_{i,j=1}^{B}\left|\left(\widehat{F}_{1,B}(X_i,X_j) - \widehat{F}_{2,B}(X_i,X_j)\right) - (F_1(X_i,X_j) - F_2(X_i,X_j))\right|\right]$$

$$= \mathbb{E}\left[\frac{1}{B^2}\sum_{i,j=1}^{B}\left|\left(\widehat{F}_{1,B}(X_i,X_j) - F_1(X_i,X_j)\right) - \left(\widehat{F}_{2,B}(X_i,X_j) - F_2(X_i,X_j)\right)\right|\right]$$

$$\leq \mathbb{E}\left[\frac{1}{B^2}\sum_{i,j=1}^{B}\left(\left|\widehat{F}_{1,B}(X_i,X_j) - F_1(X_i,X_j)\right| + \left|\widehat{F}_{2,B}(X_i,X_j) - F_2(X_i,X_j)\right|\right)\right]$$

$$= \frac{1}{B^2}\sum_{i,j=1}^{B}\left(\mathbb{E}\left[\left|\widehat{F}_{1,B}(X_i,X_j) - F_1(X_i,X_j)\right|\right] + \mathbb{E}\left[\left|\widehat{F}_{2,B}(X_i,X_j) - F_2(X_i,X_j)\right|\right]\right) .$$

***Control empirical CDF deviations.*** *For fixed $i \in \{1,2\}$ and fixed $x, x' \in \mathcal{X}$, recall that the empirical CDF is defined as:*

$$\widehat{F}_{i,B}(x,x') := \frac{1}{B}\sum_{k=1}^{B}\mathbb{1}\left\{d_i(x,X_k) \leq d_i(x,x')\right\},$$

*where $X_1, \ldots, X_B \sim \mathcal{D}$ i.i.d.*

*Each summand is a Bernoulli random variable with mean:*

$$\mathbb{E}\left[\mathbb{1}\left\{d_i(x,X_k) \leq d_i(x,x')\right\}\right] = \mathbb{P}(d_i(x,X_k) \leq d_i(x,x')) = F_i(x,x').$$

*Hence, $\widehat{F}_{i,B}(x,x')$ is the average of $B$ i.i.d. Bernoulli random variables with parameter $F_i(x,x')$, and has variance:*

$$\mathrm{Var}\left(\widehat{F}_{i,B}(x,x')\right) = \frac{1}{B}F_i(x,x')(1 - F_i(x,x')) \leq \frac{1}{4B}.$$

*By Jensen's inequality with function $t \mapsto t^2$ convex, we have:*

$$\mathbb{E}\left|\widehat{F}_{i,B}(x,x') - F_i(x,x')\right| \leq \sqrt{\mathrm{Var}(\widehat{F}_{i,B}(x,x'))} \leq \frac{1}{2\sqrt{B}}.$$

*As this hold for any $x, x'$, for each fixed $i \in \{1,2\}$, we therefore obtain*

$$E_1 \leq \frac{1}{B^2}\sum_{i,j}\left(\frac{1}{2\sqrt{B}} + \frac{1}{2\sqrt{B}}\right) = \frac{B^2}{B^2}\cdot\frac{1}{\sqrt{B}} = \frac{1}{\sqrt{B}}.$$

***Step 4: Bound $E_2$***

*Using Lemma A.1, there exists a constant $C'$ (independent of $B$) such that*

$$E_2 \leq \frac{C'}{\sqrt{B}}.$$

***Final bound:***

$$\mathbb{E}\left[\left|\frac{1}{B^2}\sum_{i,j=1}^{B}|F_1(X_i,X_j) - F_2(X_i,X_j)| - \mathrm{DC}\right|\right] \leq E_1 + E_2 \leq \frac{1+C'}{\sqrt{B}} = \frac{C}{\sqrt{B}} .$$

**Remark A.1** *No assumptions about ties or continuity of $d_i$ are needed for this expectation convergence.*

### A.1 Technical lemma

**Lemma A.1** *Let $X_1, \ldots, X_B$ be i.i.d. draws from a distribution $\mathcal{D}$ on a measurable space $\mathcal{X}$. Let $h : \mathcal{X} \times \mathcal{X} \to \mathbb{R}$ be a measurable kernel (not assumed symmetric). Define $\theta := \mathbb{E}\big[h(X_1, X_2)\big]$. Assume $h$ is uniformly bounded: $\|h\|_\infty := \sup_{x,y} |h(x,y)| =: M < \infty$. We define*

$$
V_B \;:=\; \frac{1}{B^2} \sum_{i=1}^{B} \sum_{j=1}^{B} h(X_i, X_j).
$$

*Then,*

$$
\mathbb{E}\big[\,|V_B - \theta|\,\big] \leq \frac{2(\sqrt{2} + 6\sqrt{3})M}{\sqrt{B}}.
$$

*In particular, $\mathbb{E}|V_B - \theta| = O(B^{-1/2})$.*

**Proof A.2** *The proof follows the standard Hoeffding (projection) decomposition and a counting argument for the variance of the degenerate component; no symmetry of $h$ is required because we sum over all ordered pairs $(i, j)$.*

**1. Hoeffding decomposition (ordered pairs).** *Let $X'$ denote an independent copy of $X_1$. Define the first-order projection*

$$
h_1(x) := \mathbb{E}\big[h(x, X')\big] - \theta,
$$

*and the second-order (degenerate) remainder*

$$
h_2(x, y) := h(x, y) - h_1(x) - \tilde{h}_1(y) - \theta,
$$

*where for notational symmetry we also set*

$$
\tilde{h}_1(y) := \mathbb{E}\big[h(X', y)\big] - \theta.
$$

*By construction*

$$
\mathbb{E}[h_1(X)] = \mathbb{E}[\tilde{h}_1(X)] = 0, \qquad \mathbb{E}[h_2(X, y)] = \mathbb{E}[h_2(x, X)] = 0 \quad \text{for all } x, y.
$$

*The kernel decomposes as*

$$
h(x, y) = \theta + h_1(x) + \tilde{h}_1(y) + h_2(x, y).
$$

**2. Decomposition of the V-statistic.** *Plugging this into $V_B$ gives*

$$
V_B - \theta = \frac{1}{B^2} \sum_{i,j} \Big(h_1(X_i) + \tilde{h}_1(X_j) + h_2(X_i, X_j)\Big)
$$

$$
= \frac{1}{B^2} \Big(B \sum_{i=1}^{B} h_1(X_i) + B \sum_{j=1}^{B} \tilde{h}_1(X_j)\Big) + \frac{1}{B^2} \sum_{i,j} h_2(X_i, X_j)
$$

$$
= \frac{1}{B} \sum_{i=1}^{B} h_1(X_i) + \frac{1}{B} \sum_{j=1}^{B} \tilde{h}_1(X_j) + R_B,
$$

*where we denote*

$$
R_B := \frac{1}{B^2} \sum_{i,j=1}^{B} h_2(X_i, X_j).
$$

*Define the linear part $L_B := \frac{1}{B} \sum_i h_1(X_i) + \frac{1}{B} \sum_j \tilde{h}_1(X_j)$, so $V_B - \theta = L_B + R_B$.*

**3. Bound the linear part $L_B$.** *The two sample sums are independent and mean zero; hence*

$$\operatorname{Var}(L_B) = \frac{1}{B^2} \cdot B \operatorname{Var}(h_1(X)) + \frac{1}{B^2} \cdot B \operatorname{Var}(\tilde{h}_1(X)) = \frac{1}{B}\big(\operatorname{Var}(h_1(X)) + \operatorname{Var}(\tilde{h}_1(X))\big).$$

*By Jensen,*

$$\mathbb{E}[|L_B|] \le \sqrt{\operatorname{Var}(L_B)} = \frac{1}{\sqrt{B}}\sqrt{\operatorname{Var}(h_1(X)) + \operatorname{Var}(\tilde{h}_1(X))}.$$

*It is easy to see that $|h_1| \le 2M$. Hence, $\operatorname{Var}(h_1(X)) \le 4M^2$. Similarly, $\operatorname{Var}(\tilde{h}_1(X)) \le 4M^2$. Therefore*

$$\mathbb{E}[|L_B|] \le \frac{2\sqrt{2}M}{\sqrt{B}}. \tag{1}$$

**4. Bound the degenerate part $R_B$.** *We bound $\mathbb{E}[|R_B|]$ via its second moment:*

$$\mathbb{E}[|R_B|] \le \sqrt{\mathbb{E}[R_B^2]} = \sqrt{\operatorname{Var}(R_B)}.$$

*Compute*

$$\operatorname{Var}(R_B) = \frac{1}{B^4} \sum_{i,j,k,\ell=1}^{B} \operatorname{Cov}\big(h_2(X_i, X_j),\, h_2(X_k, X_\ell)\big).$$

*Independence implies that if the index sets $\{i,j\}$ and $\{k,\ell\}$ are disjoint (i.e. all four indices are distinct), then the covariance is zero. Thus only quadruples with at least one shared index contribute. We count these quadruples.*

*The number of ordered quadruples with all four indices distinct equals*

$$B(B-1)(B-2)(B-3) = B^4 - 6B^3 + 11B^2 - 6B,$$

*so the number of quadruples with at least one common index is*

$$B^4 - \big(B^4 - 6B^3 + 11B^2 - 6B\big) = 6B^3 - 11B^2 + 6B \le 12B^3.$$

*Notice that $\mathbb{E}[h_2(X_i, X_j)] = 0$. Therefore,*

$$\big|\operatorname{Cov}\big(h_2(X_i, X_j)h_2(X_k, X_\ell)\big)\big| = \big|\mathbb{E}[h_2(X_i, X_j) \cdot h_2(X_k, X_\ell)]\big| \le \sup|h_2|^2 \,.$$

*Moreover, $|h_2| \le |h| + |h_1| + |\tilde{h}_1| + |\theta|$. It is easy to see that $|\theta| \le M$, $|h_1| \le 2M$ and $|\tilde{h}_1| \le 2M$. That is, $|h_2| \le 6M$. Therefore,*

$$\big|\operatorname{Cov}\big(h_2(X_i, X_j)h_2(X_k, X_\ell)\big)\big| \le 36M^2 \,.$$

*Consequently,*

$$\operatorname{Var}(R_B) \le \frac{12B^3}{B^4} \cdot 36M^2 = \frac{(12 \cdot 36)M^2}{B}.$$

*Therefore*

$$\mathbb{E}[|R_B|] \le \sqrt{\operatorname{Var}(R_B)} \le \frac{12\sqrt{3}\,M}{\sqrt{B}}. \tag{2}$$

**5. Combine the two bounds.** *Using the triangle inequality,*

$$\mathbb{E}\big[|V_B - \theta|\big] \le \mathbb{E}[|L_B|] + \mathbb{E}[|R_B|].$$

*Plugging (1) and (2) yields*

$$\mathbb{E}\big[|V_B - \theta|\big] \le \frac{M}{\sqrt{B}}\big(2\sqrt{2} + 12\sqrt{3}\big) = \frac{2(\sqrt{2} + 6\sqrt{3})M}{\sqrt{B}}.$$

# Appendix B    Proof of Theorems 4.2 and 4.3

## B.1    Proof of Theorem 4.2

**Theorem B.1 (Theorem 4.2 in the main text)** *Assume that $f_2$ is $\alpha$-perception coherent with $f_1$ at $x \in \mathcal{X}$ ($\phi_{f_1,f_2}(x) \geq \alpha$). Then, for all $\varepsilon_2 > \varepsilon_1 > 0$, we have*

$$\mathbb{P}_{X,X'}\left(|F_2(x,X') - F_2(x,X)| \geq \varepsilon_2 \ \Big| \ |F_1(x,X') - F_1(x,X)| \leq \varepsilon_1\right) \leq \frac{C_{\varepsilon_1}(1-\alpha)}{\varepsilon_2 - \varepsilon_1} \ , \tag{8}$$

*where $C_{\varepsilon_1}$ is a positive constant depending on $\varepsilon_1$.*

**Proof B.1** *For the sake of brevity, let $A(X,X')$ be the event $|F_1(x,X') - F_1(x,X)| \leq \varepsilon_1$. We have*

$$\mathbb{P}_{X,X'}\left(|F_2(x,X') - F_2(x,X)| \geq \varepsilon_2 \ \Big| \ A(X,X')\right)$$

$$= \mathbb{P}_{X,X'}\left(|F_2(x,X') - F_1(x,X') + F_1(x,X') - F_1(x,X) + F_1(x,X) - F_2(x,X)| \geq \varepsilon_2 \ \Big| \ A(X,X')\right)$$

$$\leq \mathbb{P}_{X,X'}\left(|F_2(x,X') - F_1(x,X')| + |F_1(x,X') - F_1(x,X)| + |F_1(x,X) - F_2(x,X)| \geq \varepsilon_2 \ \Big| \ A(X,X')\right)$$

$$\leq \mathbb{P}_{X,X'}\left(|F_2(x,X') - F_1(x,X')| + |F_1(x,X) - F_2(x,X)| \geq \varepsilon_2 - \varepsilon_1 \ \Big| \ A(X,X')\right)$$

$$\leq \mathbb{P}_{X,X'}\left(\max\left(|F_2(x,X') - F_1(x,X')|, |F_1(x,X) - F_2(x,X)|\right) \geq \frac{\varepsilon_2 - \varepsilon_1}{2} \ \Big| \ A(X,X')\right) \ .$$

*Using Lemma B.1, we have*

$$\mathbb{P}_{X,X'}\left(\max\left(|F_2(x,X') - F_1(x,X')|, |F_1(x,X) - F_2(x,X)|\right) \geq \frac{\varepsilon_2 - \varepsilon_1}{2} \ \Big| \ A(X,X')\right)$$

$$\leq \frac{4(1-\alpha)}{(\varepsilon_2 - \varepsilon_1)\mathbb{P}_{X,X'}\left(A(X,X')\right)} = \frac{C_{\varepsilon_1}(1-\alpha)}{\varepsilon_2 - \varepsilon_1} \ ,$$

*where $C_{\varepsilon_1} = \frac{4}{\mathbb{P}_{X,X'}(|F_1(x,X') - F_1(x,X)| \leq \varepsilon_1)}$ . Therefore,*

$$\mathbb{P}_{X,X'}\left(|F_2(x,X') - F_2(x,X)| \geq \varepsilon_2 \ \Big| \ A(X,X')\right) \leq \frac{C_{\varepsilon_1}(1-\alpha)}{\varepsilon_2 - \varepsilon_1} \ .$$

*This completes the proof.*

## B.2    Proof of Theorem 4.3

**Theorem B.2 (Theorem 4.3 in the main text)** *Assume that $f_2$ is $\alpha$-perception coherent with $f_1$ at $x \in \mathcal{X}$ ($\phi_{f_1,f_2}(x) \geq \alpha$). Then, for all $\varepsilon > 0$, we have*

$$\mathbb{P}_{X,X'}\left(F_2(x,X') - F_2(x,X) \geq \varepsilon \ \Big| \ F_1(x,X') \leq F_1(x,X)\right) \leq C(1-\alpha)/\varepsilon \ , \tag{9}$$

*where $C$ is a positive constant.*

**Proof B.2** *For the sake of brevity, let $A(X,X')$ be the event $F_1(x,X') \leq F_1(x,X)$. We have that*

$$\mathbb{P}_{X,X'}\left(F_2(x,X') - F_2(x,X) \geq \varepsilon \ \Big| \ A(X,X')\right)$$

$$= \mathbb{P}_{X,X'}\left(F_2(x,X') - F_2(x,X) + (F_1(x,X') - F_1(x,X)) - (F_1(x,X') - F_1(x,X)) \geq \varepsilon \ \Big| \ A(X,X')\right)$$

$$= \mathbb{P}_{X,X'}\left(F_2(x,X') - F_1(x,X') + F_1(x,X) - F_2(x,X) + F_1(x,X') - F_1(x,X) \geq \varepsilon \ \Big| \ A(X,X')\right)$$

$$\leq \mathbb{P}_{X,X'}\left(F_2(x,X') - F_1(x,X') + F_1(x,X) - F_2(x,X) \geq \varepsilon \ \Big| \ A(X,X')\right)$$

$$\leq \mathbb{P}_{X,X'}\left(\max\left(F_2(x,X') - F_1(x,X'), F_1(x,X) - F_2(x,X)\right) \geq \varepsilon/2 \ \Big| \ A(X,X')\right)$$

$$\leq \mathbb{P}_{X,X'}\left(\max\left(|F_2(x,X') - F_1(x,X')|, |F_1(x,X) - F_2(x,X)|\right) \geq \varepsilon/2 \ \Big| \ A(X,X')\right) \ .$$

*Using Lemma B.1, we have*

$$\mathbb{P}_{X,X'}\left(\max\left(\left|F_2(x,X')-F_1(x,X')\right|,\left|F_1(x,X)-F_2(x,X)\right|\right)\geq\varepsilon/2\mid A(X,X')\right)$$
$$\leq\frac{4(1-\alpha)}{\varepsilon\mathbb{P}_{X,X'}\left(A(X,X')\right)}=C(1-\alpha)/\varepsilon\ .$$

*This leads to the result of Theorem 4.3, which completes the proof.*

### B.3 Lemma B.1

**Lemma B.1** *Let $X$ and $X'$ be the i.i.d random variables following the law $\mathcal{D}_{\mathcal{X}}$ and $A(X,X')$ be an event depending on these two variables, $A$ is symmetric in $X$ and $X'$. Assume that $f_2$ is $\alpha$-perception coherent with $f_1$ at $x\in\mathcal{X}$. Then, for any $\varepsilon>0$, we have that*

$$\mathbb{P}_{X,X'}\left(\max\left(\left|F_2(x,X')-F_1(x,X')\right|,\left|F_1(x,X)-F_2(x,X)\right|\right)\geq\varepsilon\mid A(X,X')\right)$$
$$\leq\frac{2(1-\alpha)}{\varepsilon\mathbb{P}_{X,X'}\left(A(X,X')\right)}\ . \tag{10}$$

**Proof B.3** *Let $B(X,X')$ be the event $\left|F_2(x,X)-F_1(x,X)\right|\geq\left|F_1(x,X')-F_2(x,X')\right|$. We note that $B(X',X)$ is the event $\left|F_2(x,X)-F_1(x,X)\right|\leq\left|F_1(x,X')-F_2(x,X')\right|$. Furthermore, let $C(X,X')$ be the event $\max\left(\left|F_2(x,X')-F_1(x,X')\right|,\left|F_1(x,X)-F_2(x,X)\right|\right)\geq\varepsilon$.*

$$\mathbb{P}_{X,X'}\left(C(X,X')\mid A(X,X')\right)$$
$$=\mathbb{P}_{X,X'}\left(C(X,X')\bigcap B(X,X')\mid A(X,X')\right)+\mathbb{P}_{X,X'}\left(C(X,X')\bigcap\overline{B(X,X')}\mid A(X,X')\right)\ .$$

*Using Assumption 4.2, we have*

$$\mathbb{P}_{X,X'}\left(C(X,X')\bigcap\overline{B(X,X')}\mid A(X,X')\right)=\mathbb{P}_{X,X'}\left(C(X,X')\bigcap B(X',X)\mid A(X,X')\right)\ .$$

*Therefore,*

$$\mathbb{P}_{X,X'}\left(C(X,X')\mid A(X,X')\right)$$
$$=\mathbb{P}_{X,X'}\left(C(X,X')\bigcap B(X,X')\mid A(X,X')\right)+\mathbb{P}_{X,X'}\left(C(X,X')\bigcap B(X',X)\mid A(X,X')\right)$$
$$=2\mathbb{P}_{X,X'}\left(C(X,X')\bigcap B(X,X')\mid A(X,X')\right)\ (A(X,X')\ and\ C(X,X')\ symmetric\ in\ X\ and\ X')$$
$$=2\mathbb{P}_{X,X'}\left(C(X,X')\mid A(X,X')\bigcap B(X,X')\right)\mathbb{P}_{X,X'}\left(B(X,X')\mid A(X,X')\right)\ .$$

*We note that $\left(B(X,X')\mid A(X,X')\right)+\left(\overline{B(X,X')}\mid A(X,X')\right)=1$. Using Assumption 4.2, we have that $\left(B(X,X')\mid A(X,X')\right)+\left(B(X,X')\mid A(X,X')\right)=1$. As $A(X,X')$ is symmetric in $X$ and $X'$, we obtain that $2\left(B(X,X')\mid A(X,X')\right)=1$. Hence, $\left(B(X,X')\mid A(X,X')\right)=1/2$. Therefore,*

$$\mathbb{P}_{X,X'}\left(C(X,X')\mid A(X,X')\right)$$
$$=2\mathbb{P}_{X,X'}\left(C(X,X')\mid A(X,X')\bigcap B(X,X')\right)\cdot\frac{1}{2}$$
$$=\mathbb{P}_{X,X'}\left(C(X,X')\mid A(X,X')\bigcap B(X,X')\right)$$
$$=\mathbb{P}_{X,X'}\left(\left|F_1(x,X)-F_2(x,X)\right|\geq\varepsilon\mid A(X,X')\bigcap B(X,X')\right)$$
$$=\frac{\mathbb{P}_{X,X'}\left(\left|F_1(x,X)-F_2(x,X)\right|\geq\varepsilon\bigcap A(X,X')\bigcap B(X,X')\right)}{\mathbb{P}_{X,X'}\left(A(X,X')\bigcap B(X,X')\right)}$$

$$\leq \frac{\mathbb{P}_{X,X'}\left(|F_1(x,X) - F_2(x,X)| \geq \varepsilon\right)}{\mathbb{P}_{X,X'}\left(A(X,X')\bigcap B(X,X')\right)}$$

$$= \frac{\mathbb{P}_X\left(|F_1(x,X) - F_2(x,X)| \geq \varepsilon\right)}{\mathbb{P}_{X,X'}\left(A(X,X')\bigcap B(X,X')\right)} \quad \textit{(numerator independent of } X')$$

$$\leq \frac{\mathbb{E}_X\left[|F_1(x,X) - F_2(x,X)|\right]}{\varepsilon\mathbb{P}_{X,X'}\left(A(X,X')\bigcap B(X,X')\right)} \quad \textit{(Markov's inequality)}$$

$$\leq \frac{(1-\alpha)}{\varepsilon\mathbb{P}_{X,X'}\left(A(X,X')\bigcap B(X,X')\right)} \ .$$

*Note that by using Assumption 4.2 once again, we have*

$$\mathbb{P}_{X,X'}\left(A(X,X')\right) = \mathbb{P}_{X,X'}\left(A(X,X')\bigcap B(X,X')\right) + \mathbb{P}_{X,X'}\left(A(X,X')\bigcap \overline{B}(X,X')\right)$$

$$= \mathbb{P}_{X,X'}\left(A(X,X')\bigcap B(X,X')\right) + \mathbb{P}_{X,X'}\left(A(X,X')\bigcap B(X',X)\right)$$

$$= 2\mathbb{P}_{X,X'}\left(A(X,X')\bigcap B(X,X')\right) \ .$$

*Hence, we obtain*

$$\mathbb{P}_{X,X'}\left(C(X,X') \ \Big| \ A(X,X')\right) \leq \frac{2(1-\alpha)}{\varepsilon\mathbb{P}_{X,X'}\left(A(X,X')\right)} \ .$$

*This completes the proof.*

## Appendix C   Proof of the theorem on global coherence

### C.1   Proof of Theorem 4.4

**Theorem C.1 (Theorem 4.4 in the main text)** *Assume that $f_2$ is globally $\alpha$-perception coherent with $f_1$ (i.e., $\mathbb{E}_X[\phi_{f_1,f_2}(X)] \geq \alpha$). Let $X_1$, $X_2$ and $X_3$ be i.i.d. random variables following the law $\mathcal{D}_\mathcal{X}$. Then,*

1. ***Global rank preservation.*** *For all $\varepsilon_2 > \varepsilon_1 > 0$, we have*

$$\mathbb{P}_{X_1,X_2,X_3}\left(|F_2(X_1,X_2) - F_2(X_1,X_3)| \geq \varepsilon_2 \ \Big| \ |F_1(X_1,X_2) - F_1(X_1,X_3)| \leq \varepsilon_1\right)$$

$$\leq \frac{C_{\varepsilon_1}(1-\alpha)}{\varepsilon_2 - \varepsilon_1} \ ,$$

   *where $C_{\varepsilon_1}$ is a positive constant depending on $\varepsilon_1$.*

2. ***Global relative order preservation.*** *For all $\varepsilon > 0$, we have*

$$\mathbb{P}_{X_1,X_2,X_3}\left(F_2(X_1,X_2) - F_2(X_1,X_3) \geq \varepsilon \ \Big| \ F_1(X_1,X_2) \leq F_1(X_1,X_3)\right) \leq \frac{C(1-\alpha)}{\varepsilon} \ ,$$

   *where $C$ is a positive constant.*

**Proof C.1** *1. Global rank preservation.*

*Let $A(X_1,X_2,X_3)$ be the event $|F_1(X_1,X_2) - F_1(X_1,X_3)| \leq \varepsilon_1$. Let $\varepsilon = \frac{\varepsilon_2 - \varepsilon_1}{2}$. Using the same technique as in the proof of Theorem 4.2, we can obtain that*

$$\mathbb{P}_{X_1,X_2,X_3}\left(|F_2(X_1,X_2) - F_2(X_1,X_3)| \geq \varepsilon_2 \ \Big| \ A(X_1,X_2,X_3)\right)$$

$$\leq \mathbb{P}_{X_1,X_2,X_3}\left(\max\left(|F_2(X_1,X_2) - F_1(X_1,X_2)|, |F_1(X_1,X_3) - F_2(X_1,X_3)|\right) \geq \varepsilon \ \Big| \ A(X_1,X_2,X_3)\right)$$

*Applying Lemma C.1, we have that*

$$\mathbb{P}_{X_1,X_2,X_3}\left(|F_2(X_1,X_2) - F_2(X_1,X_3)| \geq \varepsilon_2 \mid A(X_1,X_2,X_3)\right)$$

$$\leq \frac{2(1-\alpha)}{\varepsilon \mathbb{P}_{X_1,X_2,X_3}(A(X_1,X_2,X_3))} = \frac{4(1-\alpha)}{(\varepsilon_2 - \varepsilon_1)\mathbb{P}_{X_1,X_2,X_3}(A(X_1,X_2,X_3))}$$

$$= \frac{4(1-\alpha)}{(\varepsilon_2 - \varepsilon_1)\mathbb{P}_{X_1,X_2,X_3}(|F_1(X_1,X_2) - F_1(X_1,X_3)| \leq \varepsilon_1)} = \frac{C_{\varepsilon_1}(1-\alpha)}{\varepsilon_2 - \varepsilon_1} ,$$

*where $C_{\varepsilon_1} = \frac{4}{\mathbb{P}_{X_1,X_2,X_3}(|F_1(X_1,X_2) - F_1(X_1,X_3)| \leq \varepsilon_1)}$. This completes the proof.*

**2. Global relative order preservation.** *Let $A(X_1,X_2,X_3)$ be the event $F_1(X_1,X_2) \leq F_1(X_1,X_3)$. Using the same technique as in the proof of Theorem 4.3, we can obtain that*

$$\mathbb{P}_{X_1,X_2,X_3}\left(F_2(X_1,X_2) - F_2(X_1,X_3) \geq \varepsilon \mid A(X_1,X_2,X_3)\right)$$

$$\leq \mathbb{P}_{X_1,X_2,X_3}\left(\max\left(|F_2(X_1,X_2) - F_1(X_1,X_2)|, |F_1(X_1,X_3) - F_2(X_1,X_3)|\right) \geq \frac{\varepsilon}{2} \mid A(X_1,X_2,X_3)\right)$$

*Applying Lemma C.1 leads to*

$$\mathbb{P}_{X_1,X_2,X_3}\left(|F_2(X_1,X_2) - F_2(X_1,X_3)| \geq \varepsilon \mid A(X_1,X_2,X_3)\right)$$

$$\leq \frac{4(1-\alpha)}{\varepsilon \mathbb{P}_{X_1,X_2,X_3}(A(X_1,X_2,X_3))}$$

$$= \frac{4(1-\alpha)}{\varepsilon \mathbb{P}_{X_1,X_2,X_3}(A(X_1,X_2,X_3))} = \frac{C(1-\alpha)}{\varepsilon} ,$$

*where $C = \frac{4}{\mathbb{P}_{X_1,X_2,X_3}(A(X_1,X_2,X_3))}$. This completes the proof.*

## C.2  Lemma C.1

**Lemma C.1** *Let $X_1$, $X_2$ and $X_3$ be the i.i.d random variables following the law $\mathcal{D}_{\mathcal{X}}$ and $A(X_1,X_2,X_3)$ be an event depending on these three variables, $A$ is symmetric in $X_2$ and $X_3$. Assume that $f_2$ is $\alpha$-perception coherent with $f_1$ ($\mathbb{E}_X[\phi_{f_1,f_2}(X)] \geq \alpha$). Then, for any $\varepsilon > 0$, we have that*

$$\mathbb{P}_{X_1,X_2,X_3}\left(\max\left(|F_2(X_1,X_2) - F_1(X_1,X_2)|, |F_1(X_1,X_3) - F_2(X_1,X_3)|\right) \geq \varepsilon \mid A(X_1,X_2,X_3)\right)$$

$$\leq \frac{2(1-\alpha)}{\varepsilon \mathbb{P}_{X_1,X_2,X_3}(A(X_1,X_2,X_3))} . \tag{11}$$

**Proof C.2** *For brevity, we shall use some compact notations. Let $B(X_1,X_2,X_3)$ be the event $|F_2(X_1,X_2) - F_1(X_1,X_2)| \geq |F_1(X_1,X_3) - F_2(X_1,X_3)|$. We note that $B(X_1,X_3,X_2)$ is the event $|F_2(X_1,X_2) - F_1(X_1,X_2)| \leq |F_1(X_1,X_3) - F_2(X_1,X_3)|$. Furthermore, let $C(X_1,X_2,X_3)$ be the event $\max\left(|F_2(X_1,X_2) - F_1(X_1,X_2)|, |F_1(X_1,X_3) - F_2(X_1,X_3)|\right) \geq \varepsilon$. We have*

$$\mathbb{P}_{X_1,X_2,X_3}\left(C(X_1,X_2,X_3) \mid A(X_1,X_2,X_3)\right)$$

$$= \mathbb{P}_{X_1,X_2,X_3}\left(C(X_1,X_2,X_3)\bigcap B(X_1,X_2,X_3) \mid A(X_1,X_2,X_3)\right)$$

$$+ \mathbb{P}_{X_1,X_2,X_3}\left(C(X_1,X_2,X_3)\bigcap \overline{B(X_1,X_2,X_3)} \mid A(X_1,X_2,X_3)\right) .$$

*Using Assumption 4.2, we have*

$$\mathbb{P}_{X_1,X_2,X_3}\left(C(X_1,X_2,X_3)\bigcap \overline{B(X_1,X_2,X_3)} \mid A(X_1,X_2,X_3)\right)$$

$$= \mathbb{P}_{X_1,X_2,X_3}\left(C(X_1,X_2,X_3)\bigcap B(X_1,X_3,X_2) \;\middle|\; A(X_1,X_2,X_3)\right) \; .$$

*Therefore,*

$$\mathbb{P}_{X_1,X_2,X_3}\left(C(X_1,X_2,X_3) \;\middle|\; A(X_1,X_2,X_3)\right)$$

$$= \mathbb{P}_{X_1,X_2,X_3}\left(C(X_1,X_2,X_3)\bigcap B(X_1,X_2,X_3) \;\middle|\; A(X_1,X_2,X_3)\right)$$

$$+ \mathbb{P}_{X_1,X_2,X_3}\left(C(X_1,X_2,X_3)\bigcap B(X_1,X_3,X_2) \;\middle|\; A(X_1,X_2,X_3)\right)$$

$$= 2\mathbb{P}_{X_1,X_2,X_3}\left(C(X_1,X_2,X_3)\bigcap B(X_1,X_2,X_3) \;\middle|\; A(X_1,X_2,X_3)\right)$$

$$= 2\mathbb{P}_{X_1,X_2,X_3}\left(C(X_1,X_2,X_3) \;\middle|\; A(X_1,X_2,X_3)\bigcap B(X_1,X_2,X_3)\right)$$

$$\times \mathbb{P}_{X_1,X_2,X_3}\left(B(X_1,X_2,X_3) \;\middle|\; A(X_1,X_2,X_3)\right)$$

$$= 2\mathbb{P}_{X_1,X_2,X_3}\left(C(X_1,X_2,X_3) \;\middle|\; A(X_1,X_2,X_3)\bigcap B(X_1,X_2,X_3)\right) \times 1/2$$

$$= \mathbb{P}_{X_1,X_2,X_3}\left(C(X_1,X_2,X_3) \;\middle|\; A(X_1,X_2,X_3)\bigcap B(X_1,X_2,X_3)\right)$$

$$= \mathbb{P}_{X_1,X_2,X_3}\left(|F_1(X_1,X_2) - F_2(X_1,X_2)| \ge \varepsilon \;\middle|\; A(X_1,X_2,X_3)\bigcap B(X_1,X_2,X_3)\right)$$

$$= \frac{\mathbb{P}_{X_1,X_2,X_3}\left(|F_1(X_1,X_2) - F_2(X_1,X_2)| \ge \varepsilon \bigcap A(X_1,X_2,X_3)\bigcap B(X_1,X_2,X_3)\right)}{\mathbb{P}_{X,X'}\left(A(X_1,X_2,X_3)\bigcap B(X_1,X_2,X_3)\right)}$$

$$\le \frac{\mathbb{P}_{X_1,X_2,X_3}\left(|F_1(X_1,X_2) - F_2(X_1,X_2)| \ge \varepsilon\right)}{\mathbb{P}_{X_1,X_2,X_3}\left(A(X_1,X_2,X_3)\bigcap B(X_1,X_2,X_3)\right)}$$

$$= \frac{\mathbb{P}_{X_1,X_2}\left(|F_1(X_1,X_2) - F_2(X_1,X_2)| \ge \varepsilon\right)}{\mathbb{P}_{X_1,X_2,X_3}\left(A(X_1,X_2,X_3)\bigcap B(X_1,X_2,X_3)\right)} \quad \textit{(numerator independent of } X_3\textit{)}$$

$$= \frac{\mathbb{E}_{X_1,X_2}\left[\mathbb{1}_{\{|F_1(X_1,X_2)-F_2(X_1,X_2)|\ge\varepsilon\}}\right]}{\mathbb{P}_{X_1,X_2,X_3}\left(A(X_1,X_2,X_3)\bigcap B(X_1,X_2,X_3)\right)}$$

$$= \frac{\mathbb{E}_{X_1}\left[\mathbb{E}_{X_2}\left[\mathbb{1}_{\{|F_1(X_1,X_2)-F_2(X_1,X_2)|\ge\varepsilon\}}\right]\right]}{\mathbb{P}_{X_1,X_2,X_3}\left(A(X_1,X_2,X_3)\bigcap B(X_1,X_2,X_3)\right)}$$

$$= \frac{\mathbb{E}_{X_1}\left[\mathbb{P}_{X_2}\left(|F_1(X_1,X_2) - F_2(X_1,X_2)| \ge \varepsilon\right)\right]}{\mathbb{P}_{X_1,X_2,X_3}\left(A(X_1,X_2,X_3)\bigcap B(X_1,X_2,X_3)\right)}$$

$$\le \frac{\mathbb{E}_{X_1}\left[\frac{\mathbb{E}_{X_2}[|F_1(X_1,X_2)-F_2(X_1,X_2)|]}{\varepsilon}\right]}{\mathbb{P}_{X_1,X_2,X_3}\left(A(X_1,X_2,X_3)\bigcap B(X_1,X_2,X_3)\right)} \quad \textit{(Markov's inequality)}$$

$$= \frac{\mathbb{E}_{X_1}\left[1 - \phi_{f_1,f_2}(X_1)\right]}{\varepsilon\mathbb{P}_{X_1,X_2,X_3}\left(A(X_1,X_2,X_3)\bigcap B(X_1,X_2,X_3)\right)}$$

$$= \frac{1 - \mathbb{E}_{X_1}\left[\phi_{f_1,f_2}(X_1)\right]}{\varepsilon\mathbb{P}_{X_1,X_2,X_3}\left(A(X_1,X_2,X_3)\bigcap B(X_1,X_2,X_3)\right)}$$

$$\le \frac{1 - \alpha}{\varepsilon\mathbb{P}_{X_1,X_2,X_3}\left(A(X_1,X_2,X_3)\bigcap B(X_1,X_2,X_3)\right)}$$

$$= \frac{2(1 - \alpha)}{\varepsilon\mathbb{P}_{X_1,X_2,X_3}\left(A(X_1,X_2,X_3)\right)} \; .$$

*This completes the proof.*

## Appendix D  Proof of Theorem 4.5

**Theorem D.1 (Theorem 4.5 in the main text)** *Consider a non-empty set $\mathcal{A} \subseteq \mathcal{X}$ such that $\phi_{f_1,f_2}(x) \geq \alpha$, $\forall x \in \mathcal{A}$. For $0 < \delta \leq 1$, let $\mathcal{A}_\delta = \{x \in \mathcal{X} : \min_{x' \in \mathcal{A}} F(x', x) \leq \delta\}$. Then, for any $\varepsilon$ ($0 < \varepsilon \leq \alpha$), we have*

$$\mathbb{P}\left(\phi_{f_1,f_2}(X) \leq \alpha - \varepsilon \;\Big|\; X \in \mathcal{A}_\delta\right) \leq \frac{O(\delta)}{\varepsilon C_{\mathcal{A}_\delta}} \;,$$

*where $C_{\mathcal{A}_\delta}$ is a positive constant depending on $\mathcal{A}_\delta$.*

**Proof D.1** *Let us consider an $x \in \mathcal{A}_\delta$ and let $a(x)$ be an $\arg\min_{x' \in \mathcal{A}} F(x', x)$ (if not unique). We have that*

$$F_1(a(x), x) \leq \delta \text{ and } F_2(a(x), x) \leq \delta \;,$$

*which implies $d_1(x, a(x)) = O(\delta)$ and $d_2(x, a(x)) = O(\delta)$. Notice that $O(\delta)$ is uniform on $x \in \mathcal{A}_\delta$ and $x' \in \mathcal{X}$. Moreover, for any $x' \in \mathcal{X}$, we have*

$$d_1(a(x), x') - d_1(x, a(x)) \leq d_1(x, x') \leq d_1(a(x), x') + d_1(x, a(x)) \;.$$

*Hence,*

$$\begin{aligned}
F_1(x, x') &\leq \mathbb{P}_X\left(d_1(x, X) \leq d_1(a(x), x') + d_1(x, a(x))\right) \\
&= \mathbb{P}_X\left(d_1(x, X) \leq d_1(a(x), x')\right) + \mathbb{P}_X\left(d_1(a(x), x') \leq d_1(x, X) \leq d_1(a(x), x') + d_1(x, a(x))\right) \\
&= F_1(a(x), x') + O(d_1(x, a(x)) = F_1(a(x), x') + O(\delta) \;.
\end{aligned}$$

*On the other hand,*

$$\begin{aligned}
F_1(x, x') &\geq \mathbb{P}_X\left(d_1(x, X) \leq d_1(a(x), x') - d_1(x, a(x))\right) \\
&= \mathbb{P}_X\left(d_1(x, X) \leq d_1(a(x), x')\right) - \mathbb{P}_X\left(d_1(a(x), x') - d_1(x, a(x)) \leq d_1(x, X) \leq d_1(a(x), x')\right) \\
&= F_1(a(x), x') - O(d_1(x, a(x)) = F_1(a(x), x') - O(\delta) \;.
\end{aligned}$$

*Therefore, $|F_1(x, x') - F_1(a(x), x')| = O(\delta)$, for any $x' \in \mathcal{X}$. Proceeding in the same way, we also have $|F_2(x, x') - F_2(a(x), x')| = O(\delta)$. Thus,*

$$\begin{aligned}
&|F_1(x, x') - F_2(x, x')| \\
&= |F_1(x, x') - F_1(a(x), x') + F_2(a(x), x') - F_2(x, x') + F_1(a(x), x') - F_2(a(x), x')| \\
&\leq |F_1(x, x') - F_1(a(x), x')| + |F_2(a(x), x') - F_2(x, x')| + |F_1(a(x), x') - F_2(a(x), x')| \\
&= |F_1(a(x), x') - F_2(a(x), x')| + O(\delta) \;.
\end{aligned}$$

*Therefore, there exists a positive constant $C_\delta = O(\delta)$ such that $|F_1(x, x') - F_2(x, x')| \leq |F_1(a(x), x') - F_2(a(x), x')| + C_\delta$, for any $(x, x') \in \mathcal{A}_\delta \times \mathcal{X}$. Consequently,*

$$\begin{aligned}
&\mathbb{P}_X\left(\phi_{f_1,f_2}(X) \leq \varepsilon \;\Big|\; X \in \mathcal{A}_\delta\right) \\
&= \mathbb{P}_X\left(1 - \mathbb{E}_{X'}\left[|F_1(X, X') - F_2(X, X')|\right] \leq \alpha - \varepsilon \;\Big|\; X \in \mathcal{A}_\delta\right) \\
&= \mathbb{P}_X\left(\mathbb{E}_{X'}\left[|F_1(X, X') - F_2(X, X')|\right] \geq 1 - \alpha + \varepsilon \;\Big|\; X \in \mathcal{A}_\delta\right) \\
&\leq \mathbb{P}_X\left(\mathbb{E}_{X'}\left[|F_1(a(X), X') - F_2(a(X), X')|\right] + C_\delta \geq 1 - \alpha + \varepsilon \;\Big|\; X \in \mathcal{A}_\delta\right) \\
&= \frac{\mathbb{P}_X\left(\mathbb{E}_{X'}\left[|F_1(a(X), X') - F_2(a(X), X')|\right] + C_\delta \geq 1 - \alpha + \varepsilon \bigcap X \in \mathcal{A}_\delta\right)}{\mathbb{P}_X\left(X \in \mathcal{A}_\delta\right)} \\
&\leq \frac{\mathbb{P}_X\left(\mathbb{E}_{X'}\left[|F_1(a(X), X') - F_2(a(X), X')|\right] + C_\delta \geq 1 - \alpha + \varepsilon\right)}{\mathbb{P}_X\left(X \in \mathcal{A}_\delta\right)} \\
&= \frac{\mathbb{P}_X\left(1 - \phi_{f_1,f_2}(a(X)) + C_\delta \geq 1 - \alpha + \varepsilon\right)}{\mathbb{P}_X\left(X \in \mathcal{A}_\delta\right)}
\end{aligned}$$

$$\leq \frac{\mathbb{P}_X\left(1 - \alpha + C_\delta \geq 1 - \alpha + \varepsilon\right)}{\mathbb{P}_X\left(X \in \mathcal{A}_\delta\right)} \left(\phi_{f_1,f_2}(a(X)) \geq \alpha\right)$$

$$= \frac{\mathbb{P}_X\left(C_\delta \geq \varepsilon\right)}{\mathbb{P}_X\left(X \in \mathcal{A}_\delta\right)} = \frac{\mathbb{1}_{\{C_\delta \geq \varepsilon\}}}{\mathbb{P}_X\left(X \in \mathcal{A}_\delta\right)}$$

$$\leq \frac{C_\delta}{\varepsilon \mathbb{P}_X\left(X \in \mathcal{A}_\delta\right)} \left(\mathbb{1}_{\{C_\delta \geq \varepsilon\}} \leq \frac{C_\delta}{\varepsilon}\right)$$

$$= \frac{O(\delta)}{\varepsilon \mathbb{P}_X\left(X \in \mathcal{A}_\delta\right)}$$

$$\leq \frac{O(\delta)}{\varepsilon \mathbb{P}_X\left(X \in \mathcal{A}\right)} = \frac{O(\delta)}{\varepsilon C_\mathcal{A}} \ .$$

*This completes the proof of Theorem 4.5.*

## Appendix E   Experiment details

For toy datasets on 2D and 3D, we simply use Euclidean distance as the dissimilarity metric. For all the experiments on neural networks, we use the cosine dissimilarity defined as

$$d(u, v) = \frac{1}{2}\left(1 - \frac{\langle u, v \rangle}{\|u\| \cdot \|v\|}\right) \ . \tag{12}$$

All the experiments are conducted using a single GPU NVIDIA GeForce RTX 4090. The running time for each experiment is reported in each sub-section.

### E.1   Toy dataset on 2D and 3D

For these experiments, we use scikit-learn package in Python to simulate the datasets. The two-moon dataset has 700 points, and the 5 $2D$ Gaussian clusters have 1000 points. The 3D clusters also have 1000 points. The student configuration is randomly initialized using Gaussian distribution. For training, we use batch size equal to 64, and learning rate equal to 0.1. We use Adam optimizer of the Pytorch package for training during 800 epochs. We set the temperature equal to 0.1 in the soft ranking function.

### E.2   Stylized Empirical Study of Perception Coherence and Downstream Classification

For simplicity, the teacher and student models share the same architecture, consisting of a feature extractor followed by a classification head (softmax layer). The feature extractor has the following structure $\text{Linear}(2, 20) \rightarrow \text{ReLU} \rightarrow \text{Linear}(20, 20)$. The classification head (softmax layer) consists of a linear layer $\text{Linear}(20, 2)$, which maps the extracted features to two output classes, followed by a softmax function.

The teacher model is first trained with the training set (with binary labels) for 200 epochs using the *Adam* optimizer with a learning rate of $10^{-3}$. Knowledge transfer is then performed by optimizing the perception coherence of the student model to match the feature structure produced by the teacher's feature extractor. This is done by using the same training set (no label). Note that during this transfer stage, only the student feature extractor is optimized, while the classification head is excluded. The student model is trained using the *Adam* optimizer with a learning rate of $10^{-3}$ for 40 epochs. We save the student model every 4 epochs, resulting in a sequence of checkpoints $\{S_k\}_{k=1}^{10}$.

For each checkpoint $\{S_k\}_{k=1}^{10}$, we freeze the feature extractor and train a new classification head in a supervised manner using the training set (with binary labels). This supervised training is conducted for 20 epochs with the *Adam* optimizer and a learning rate of $10^{-3}$. Finally, using the test dataset, the test accuracy after this supervised training is recorded for each checkpoint, yielding $\{\text{Acc}_k\}_{k=1}^{10}$.

### E.3 Retrieval experiments on CIFAR10 and CUB-200

Following Passalis et al. (2020a), we use ResNet18 as the teacher model for both datasets. We also use the same student models for each datasets, shown in Fig. 10. We apply our method only on the penultimate layer (dim 64 in Fig. 10), without using the final classification layer.

**Training teacher model on CIFAR10.** We use batch size equal to 64. We apply during training the data augmentation techniques, including random cropping (32x32, padding=4) and random horizontal flipping. We use the optimizer with following hyper-parameters: SGD(lr=0.1, momentum=0.9, nesterov=True, weight_decay=5e-4). We train the model from scratch for 100 epochs, where we decay the learning rate by a factor of 0.2 at epochs 40 and 80. The training time is less than 30 minutes.

**Training teacher model on on the subset of CUB-200.** We use batch size equal to 64. All images are resized to 256x256. We apply during training the data augmentation techniques, including random cropping (224x224), random horizontal flipping and random rotation (30 degrees). We use the optimizer with following hyper-parameters: SGD(lr=0.001, momentum=0.9, nesterov=True, weight_decay=5e-4). We train the model from the default pretrained model of Pytorch for 100 epochs, where we decay the learning rate by a factor of 0.1 at epoch 50. The training time is less than 5 minutes.

**Transferring process.** We use the training set as the transfer set, without using label. For both datasets, we train for 150 epochs. The technique is applied on the penultimate layer of teacher model (dimension of 512) and the penultimate layer of student models (dimension of 64). For temperature, we fix $\tau = 0.1$ for teacher model and $\tau = 0.3$ for student models. Notice that we use only the training set without label. We use the optimizer as follows: SGD(lr=0.05, momentum=0.9, nesterov=True, weight_decay=5e-4). We decay the learning rate by factor 0.1 at epochs 50 and 100. For each dataset, during transferring, we apply the same data augmentation techniques as for training the teacher model. The training time for CIFAR10 is about 20 minutes. The training time for the subset of CUB200 is less than 15 minutes.

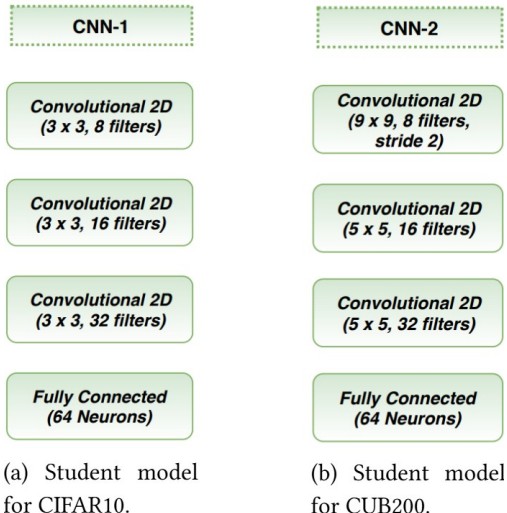

(a) Student model for CIFAR10.

(b) Student model for CUB200.

Figure 10: Student models for CIFAR10 and CUB200. Images are extracted from Passalis et al. (2020a). Notice that we perform transferring on the features of penultimate layer (dimension of 64), and do not use the final classification layer.

### E.4 Classification on CIFAR-100

For teacher models, we use pretrained models released in the official code of the work Tian et al. (2020) without any modification. We also use the same student models from the same code. We follow the same training scheme as the work Tian et al. (2020). More precisely, for optimizer, we use SGD(lr=0.01, momentum=0.9, weight_decay=5e-4). We use the training set for transferring, during 240 epochs, where we decay the learning by a factor of 0.1 at epochs 150, 180 and 210. The batch size is 64. Data augmentation

techniques include random cropping (32x32, padding=4) and random horizontal flipping. The most recent VRM method (Zhang et al., 2025) introduces virtual relation matching and applies RandAug (Random Augmentation). For a fair comparison, in addition to random cropping and random horizontal flipping, we also add RandAug to the inputs using the official code associated with this last work. We fix temperature $\tau = 0.2$ for teacher model and $\tau = 0.3$ for student models. Our technique is applied on the penultimate layer and the logit layer (before softmax function). The total loss for training the classification student model is $\mathcal{L}_{\text{total}} = \mathcal{L}_{\text{cross-entropy}} + \lambda \mathcal{L}_{\text{ours}}$, where we fix $\lambda = 5$. We perform 5 independent runs to report results in Table 4. The running time for one run (240 epochs) is as follows: ShuffleNetV1: about 45 minutes. ShuffleNetV2: about 52 minutes. MobileNetV2: about 1 hour.

### E.5 Ablation study: effect of scaling temperature in the soft ranking

In this section, we investigate the effect of scaling temperature in the soft ranking. We fix the temperature of the teach models as precised in Section E.3, and use different temperatures for the student model. Results are shown in Table 7.

Table 7: Ablation study: performance model with different values of $\tau$.

| $\tau$ (temperature) | | 0.001 | 0.01 | 0.05 | 0.1 | 0.3 | 0.5 | 5.0 | 10.0 |
|---|---|---|---|---|---|---|---|---|---|
| CUB200 subset | mAP | 15.61 | 16.93 | 14.30 | 17.24 | 28.42 | 30.38 | 14.16 | 13.17 |
| | top-10 | 22.67 | 24.29 | 20.11 | 23.86 | 36.55 | 38.49 | 18.87 | 18.04 |
| CIFAR10 | mAP | 50.22 | 49.36 | 49.50 | 50.86 | 54.25 | 53.72 | 27.38 | 25.14 |
| | top-100 | 61.02 | 60.27 | 60.22 | 61.36 | 65.00 | 63.89 | 32.43 | 28.51 |

From Table 7, we have the following remarks:

- **Small Temperature.** With very small $\tau$, the sigmoid approaches a step function, causing vanishing gradients almost everywhere. This was confirmed in our ablation study on CUB200, where using $\tau \in \{0.001, 0.01, 0.05\}$ consistently led to degraded performance.

- **Large Temperature.** Conversely, large $\tau$ (e.g., $\tau \geq 5$) causes the sigmoid $\Lambda$ to behave nearly linearly, with: $(x) \approx \frac{1}{2} + \frac{x}{4\tau}$ for small $|x/\tau|$. This weakens the ranking effect and makes the method rely more on raw dissimilarity magnitudes, which is suboptimal in lightweight settings (e.g., when $f_1 \ll f_2$ in capacity). We observed performance drops in both CIFAR100 and CUB200 in this case.

- **Summary.** In practice, intermediate values $\tau \in \{0.1, 0.3, 0.5\}$ yield the best results. This suggests that further temperature tuning might be useful to have optimal results.

## Appendix F  Computational Complexity Comparison

In this section, we compare the computational complexity of different teacher-student pairs in the experiments. As required in practical deployment scenarios, all student models are strictly smaller and computationally cheaper than their corresponding teacher models, with details reports in Table 8. These results are obtained using the python package **ptflops** (Sovrasov, 2018-2024).

Table 8: Computational complexity comparison (in MACs) of teacher-student pairs. **MAC** denotes multiply-accumulate operations, where 1 MAC $\approx$ 2 FLOPs (one multiplication and one addition).

| | CIFAR10 | SVHN | CIFAR100 (ResNet-50 → MobileNetV2) | CIFAR100 (ResNet-32x4 → ShuffleNetV1) | CIFAR100 (ResNet-32x4 → ShuffleNetV2) |
|---|---|---|---|---|---|
| **Teacher model** | 557.22 MMAC | 1.82 GMAC | 1.31 GMAC | 1.09 GMAC | 1.09 GMAC |
| **Student model** | 523.15 KMAC | 37.26 MMAC | 7.35 MMAC | 41.58 MMAC | 46.06 MMAC |

