# OpenReview forum: "Feature Representation Transferring to Lightweight Models via Perception Coherence"
_TMLR — Accepted by TMLR_

### Review · Reviewer_N8ii · 2025-10-24

**Summary Of Contributions:**

This paper proposes a new knowledge distillation (KD) method for transferring feature representations from a large teacher model to a smaller student model. The main contributions are as follows:

1. Introduction of “Perception Coherence”: The authors present a novel framework emphasizing consistency in how teacher and student models perceive data. Rather than forcing the student to replicate the teacher’s exact geometry, the method aligns their relative rankings of pairwise dissimilarities across data points, offering a more flexible and realistic goal for low-capacity students.

2. Differentiable Loss Design: The paper introduces a differentiable ranking loss based on soft sigmoid approximations, enabling gradient-based optimization over inherently non-differentiable ranking operations.

3. Theoretical and Empirical Validation: The authors provide theoretical guarantees for convergence and demonstrate the relationship between perception coherence and order-preserving probability. Experiments on toy data, image retrieval, and classification tasks show consistent improvements over prior KD methods such as PKT and CRD.


__Strengths__

- Novel and Intuitive Concept: Shifting from absolute distance alignment to relative ranking preservation is both practical and well-motivated, especially for lightweight models with limited representational capacity.

- Strong Theoretical Foundation: The paper rigorously analyzes the mathematical properties of the proposed framework, linking theory to empirical behavior and increasing the credibility of the approach.


__Weaknesses__

- Computational Cost for Lightweight Models:
The loss computation involves all pairwise dissimilarities within a mini-batch, resulting in
 $O(B^2)$ complexity. For lightweight student models, this can dominate training cost, potentially becoming the main bottleneck. Unlike contrastive learning—where large models and distributed setups justify this cost—here the mismatch between method complexity and target application (efficient model training) warrants deeper discussion.

- Missing Comparisons with Recent Work:
The experimental section lacks comparisons with post-2020 KD methods such as ReviewKD (CVPR 2021) and SimKD (CVPR 2022). These methods introduce high-order or classifier-reused relational structures, conceptually close to the proposed idea. Including at least ReviewKD or SimKD in classification benchmarks would better situate this work in the current literature.

__Reference__
- [ReviewKD] https://arxiv.org/pdf/2104.09044
- [SimKD] https://arxiv.org/abs/2203.14001

**Additional Comments:**

This is a well-written and theoretically solid paper introducing a creative and meaningful idea. With stronger empirical grounding through up-to-date comparisons and clearer discussion of computational trade-offs, the paper could make a significant contribution to the KD literature.

**Audience:**

Yes

**Audience Explanation:**

The paper’s theoretical framework and practical method for knowledge distillation would interest readers in TMLR who focus on representation learning, model compression, and efficient deep learning.

**Broader Impact Concerns:**

The paper addresses model compression and efficiency rather than application-level AI behavior. No ethical concerns are apparent. On the contrary, improving lightweight model training can reduce computational demands and broaden AI accessibility on edge devices.

**Claims And Evidence:**

No

**Claims Explanation:**

The main claim that preserving relative perceptual order enables effective representation transfer is theoretically sound, but the empirical support could be stronger.

- Theoretical Evidence: Section 4 provides clear guarantees, e.g., the estimation error of perception coherence decreases with the square root of batch size (Theorem 4.1), and higher coherence correlates with better order preservation (Theorems 4.2–4.4).
- Empirical Evidence: Figures 4–5 effectively visualize structural transfer on toy data. Tables 1–2 show clear mAP improvements over PKT and HKD, confirming stronger feature learning. The only gap is the absence of comparisons with newer baselines in classification tasks.

__Main Concern__
One concern is the lack of comparison with methods developed after 2020. The authors’ empirical evidence is limited to comparisons with studies published before 2020 and does not sufficiently capture the progress made in the field over the past five years.

**Requested Changes:**

While the contribution of this paper theoretically sound, the empirical validation and comparison with recent related work are somewhat limited. The following revisions would substantially strengthen the paper.

__Clarify the role of the hyperparameter τ (temperature) in soft ranking:__
The appendix currently includes a discussion on τ, but a concise summary should also appear in the main text. In particular, please explain why different τ values are used for the teacher and student models—e.g., whether this relates to differences in feature variance or prediction confidence. This clarification would help readers understand the method more deeply and facilitate reproducibility.

__Expand comparative experiments with recent related work:__
As noted in the Weaknesses section, please consider adding experimental comparisons in the classification task with more recent KD methods, at least ReviewKD (CVPR 2021) and SimKD (CVPR 2022). This would allow the proposed method’s advantages to be validated in a contemporary context and further enhance the impact of the paper.
For reference, VRM: Knowledge Distillation via Virtual Relation Matching (ICCV 2025, the first version (v1) of the preprint was uploaded at the end of February 2025, and the third version (v3) was uploaded at the end of July.)  provides a comprehensive overview of comparable relational KD approaches and may serve as a useful benchmark.

__Minor Comment:__
Some experimental plots are not properly embedded and are difficult to read (e.g., Fig. 7).

__Reference__
- [ReviewKD] https://arxiv.org/abs/2104.09044
- [SimKD] https://arxiv.org/abs/2203.14001
- [VRM] https://arxiv.org/abs/2502.20760

---

> ### Author Response · Authors · 2025-10-30
> **Response of the authors to the review**
>
> First of all, thank you for your kindly insightful reviews. Here are our answer. Many aspects of the discussion will be added to the paper.
>
> ## Computational Cost for Lightweight Models
>
> "The loss computation involves all pairwise dissimilarities within a mini-batch... Unlike contrastive learning—where large models and distributed setups justify this cost—here the mismatch between method complexity and target application (efficient model training) warrants deeper discussion."
>
> In fact, we should distinguish **training** from **inference**.
> In many applications, the lightweight model is trained on a server (or a machine with sufficient computational resources). Once trained, the model is **deployed to the device** for inference. As a matter of fact, this setup also matches our **industrial context**, where we train models on computers and then embed the trained models into devices for real-world use. So, the training cost does not really impact the lightweight model's deployment.
>
> We find this reviewer’s remark insightful, and we will add this clarification to the discussion section for better context.
>
> ---
>
> ## Clarify the Role of the Hyperparameter τ (Temperature) in Soft Ranking
>
> - **Concise summary in the main text:**
>   Yes, we will add a concise explanation of the temperature parameter τ in the main text, with reference to the detailed discussion in the appendix.
>
> - **Different τ values for teacher and student models:**
>   This mainly relates to training dynamics. We observe that using a slightly **larger τ for the student** facilitates learning, likely because a too small  τ can weaken gradient signals during backpropagation.
>   Conversely, a **smaller τ for the teacher** helps it focus more on relative ranking rather than absolute distance magnitudes, since a larger τ makes the sigmoid function more linear.
>
>   In our experiments, we intentionally kept the **same τ value** across different teacher–student pairs for simplicity. However, further tuning may yield better results.
>   In practice, good values (for both teacher and student) are typically in the range **{0.1, 0.2, 0.3}**.
>   While we do not yet have a general formula for the optimal τ, this practical range should be useful for practitioners applying our method to their own tasks.
>
>   We will add a new **“Practical Aspects”** section for this.
>   Thank you for this helpful suggestion.
>
> ---
>
> ## Expand Comparative Experiments with Recent Related Work
>
> Following your recommendation, we have added comparisons with **recent state-of-the-art methods**, including **VRM (ICCV 2025)**, **ReviewKD (CVPR 2021)**, **DIST (NeurIPS 2022)**, and **TTM (ICLR 2024)**.
>
> The most recent **VRM** method introduces *virtual relation matching* and applies *RandAug* (Random Augmentation).
> For a fair comparison, we added RandAug to our inputs (using the official code released with their paper) and applied our method under the same setting. The updated results are shown in the table below.
>
> We observe that our method achieves **on-par performance** compared to VRM, even though VRM includes additional components such as **inter-class relation modeling** and **edge pruning strategies**.
> Our approach intentionally uses a minimal setting, yet still matches VRM performance.
>
> This result demonstrates the effectiveness of our method.
> We appreciate this suggestion, as these additional comparisons have further strengthened our paper.
>
>
> ### Comparison of Knowledge Distillation Methods
>
> | **Method** | **ResNet-50 → MobileNetV2** | **ResNet-32x4 → ShuffleNetV1** | **ResNet-32x4 → ShuffleNetV2** |
> |---|---:|---:|---:|
> | **Teacher** | 79.34 | 79.42 | 79.42 |
> | **Student** | 64.6 | 70.5 | 71.82 |
> | **KD (Hinton-arXiv’15)** | 67.35 ± 0.32 | 74.07 ± 0.19 | 74.45 ± 0.27 |
> | **FitNet (ICLR’15)** | 63.16 ± 0.47 | 73.59 ± 0.15 | 73.54 ± 0.22 |
> | **VID (CVPR’19)** | 67.57 ± 0.28 | 73.38 ± 0.09 | 73.40 ± 0.17 |
> | **RKD (CVPR’19)** | 64.43 ± 0.42 | 72.28 ± 0.39 | 73.21 ± 0.28 |
> | **PKT (ECCV’18)** | 66.52 ± 0.33 | 74.10 ± 0.25 | 74.69 ± 0.34 |
> | **CRD (ICLR’20)** | 69.11 ± 0.36 | 75.11 ± 0.32 | 75.65 ± 0.10 |
> | **ReviewKD (CVPR’21)** | 69.89 | 77.45 | 77.78 |
> | **DIST (NeurIPS’22)** | 68.66 ± 0.23 | 76.34 ± 0.18 | 77.35 ± 0.25 |
> | **TTM (ICLR’24)** | 69.59 | 74.37 | 76.55 |
> | **VRM (with RandAug) (ICCV’25)** | **72.30** |  $\underline{78.28}$ | **79.34** |
> | **Ours (with RandAug)** | $\underline{71.10}$ ± 0.10 | **78.59** ± 0.13 |  $\underline{79.14}$ ± 0.10 |

---

### Review · Reviewer_Uj7N · 2025-10-27

**Summary Of Contributions:**

The paper introduces “perception coherence,” a probabilistic notion of agreement between a teacher’s and a student’s feature spaces. Instead of matching feature magnitudes or full geometries, the student is trained to preserve the teacher’s **relative dissimilarity rankings** around each reference point. The authors formalise this via cumulative functions and define a pointwise and global coherence level that measures how close these distributions are across inputs. They propose a differentiable, mini-batch loss that approximates ranking with a soft pairwise sigmoid and minimises the squared distance between teacher and student rank vectors (Eq. (3)–(4)). Theoretical results include an (O(B^{-1/2})) convergence rate for the mini-batch estimator of their discrepancy measure.

**Audience:**

Yes

**Audience Explanation:**

Representation transfer and knowledge distillation for small models are active topics. A probabilistic, rank-based objective that handles different feature dimensions is of clear interest to readers who work on compression, retrieval, and cross-architecture transfer. The paper is relevant to practical settings where labels are not available for the transfer set.

**Claims And Evidence:**

Yes

**Claims Explanation:**

The theory supports the estimator properties and gives qualitative guarantees about rank preservation at both local and global levels. The proofs and derivations match the claims. The experiments cover both retrieval and classification, compare with strong baselines, and include ablations that align with the theoretical signals.

**Requested Changes:**

First, add a task-level connection between coherence and error. Even a stylised result (for example, a 1-NN retrieval or margin-based classification bound under separability) would make the metric operational for downstream evaluation.

Second, discuss time and memory complexity. The loss touches many pairwise comparisons per reference.

Third, give practical guidance for the soft-ranking temperature. The ablation shows a narrow good range; a simple data-driven rule (for example, set temperature to a constant multiple of the median absolute deviation of within-batch dissimilarities) would help adoption.

---

> ### Author Response · Authors · 2025-10-30
> **Response of the authors**
>
> First of all, the author would like to thank the reviewer for the thoughtful remarks. Many aspects of the discussion will be incorporated into the revised paper.
>
> ##  Adding a Task-Level Connection Between Coherence and Error
>
> > *“Even a stylised result (for example, a 1-NN retrieval or margin-based classification bound under separability) would make the metric operational for downstream evaluation.”*
>
> Following the reviewer’s suggestion, we conducted a controlled experiment to investigate the connection between **perception coherence (PC)** and **downstream classification performance**.
>
> - **Dataset:** We use the *two-moon* dataset (2 classes) with 800 points, randomly split into 400 for training and 400 for validation.
> - **Teacher model:** A neural network with two hidden layers and a final softmax layer is trained for classification. Once trained, it achieves perfect accuracy on both the training and validation sets.
> - **Student model:** For simplicity, the student network has the same architecture as the teacher.
> - **Unsupervised transfer:** Using our proposed method, we transfer the feature representation from the *penultimate layer* (i.e., right before the softmax) of the teacher to the student. The transfer is performed using the **unlabeled training set** only. During training, we save the student model at 10 different epochs, denoted as $\{m_1, m_2, \dots, m_{10}\}$. The final softmax layer of the student is *not involved* in this transfer process.
> - **Evaluating perception coherence:** For each saved model $\{m_1, \dots, m_{10}\}$, we compute the **coherence coefficient** on the training set, yielding values $\{pc_1, pc_2, \dots, pc_{10}\}$.
> - **Downstream classification:** For each student model, we then train **only the final softmax layer** (with all other layers frozen) for supervised classification on the two-moon dataset. This allows us to assess whether the transferred, unsupervised features are useful for classification. We obtain the corresponding validation accuracies $\{acc_1, acc_2, \dots, acc_{10}\}$.
> - **Relation between PC and accuracy:** The results are summarized in the table below. We observe a strong positive correlation between training perception coherence and downstream classification accuracy: **better coherence of the learned representation leads to better performance**. The Pearson correlation coefficient is **0.920**. A scatter plot illustrating this relationship is also included in the paper.
>
> | Training perception coherence level | 0.841 | 0.877 | 0.889 | 0.901 | 0.911 | 0.921 | 0.931 | 0.940 | 0.949 | 0.956 |
> |-----------|-------|-------|-------|-------|-------|-------|-------|-------|-------|-------|
> | Downstream classification accuracy (%) | 82.00 | 83.50 | 83.25 | 85.75 | 86.75 | 88.00 | 86.75 | 89.25 | 87.00 | 90.00 |
>
> ## Discussing Computational Complexity
>
> While the theoretical computational complexity of our method is $\mathcal{O}(B^3)$ per batch—due to the need to compute and compare all pairwise distances when soft-ranking dissimilarities—the entire pipeline is **highly parallelizable**.
>
> The core operations (distance matrix computation, sigmoid-based soft ranking, and squared loss) are implemented as **batched tensor operations**, which are efficiently handled by modern deep learning libraries on **GPUs**.
>
> Importantly, since the soft ranking for each reference point (anchor point) is **independent** of others, we can **distribute the computation across multiple GPUs** by partitioning the batch along the reference point dimension. With $N$ GPUs, each device processes approximately $\frac{B}{N}$ reference points in parallel.
>
> The final loss is then aggregated using standard reduction operations (e.g., **all-reduce**), enabling **near-linear scaling** and significantly reducing the overall wall-clock training time in practice.
>
> ---
>
> ## Practical Guidance for the Soft-Ranking Temperature
>
> In practice, good values (for both teacher and student) are typically in the range **{0.1, 0.2, 0.3}**.
> While we do not yet have a general formula for the optimal $\tau$, this practical range should be useful for practitioners applying our method to their own tasks.
>
> We also observe that using a slightly larger $\tau$ for the **student** facilitates learning, likely because a too small $\tau$ weakens gradient signals during backpropagation, negatively affecting training dynamics.
> Conversely, a smaller $\tau$ for the **teacher** helps it focus more on **relative ranking** rather than absolute distance magnitudes, since a larger $\tau$ makes the sigmoid function more linear.
>
> > **Note:** In our experiments, we intentionally kept the same $\tau$ value across different teacher–student pairs for simplicity. However, further tuning may yield even better results.
>
> We will add a new **“Practical Aspects”** section to include this discussion.
> Thank you for this helpful suggestion.

---

> ### Comment · Action_Editor_ezqL · 2025-12-25
> **final recommendation**
>
> Dear reviewer Uj7N,
>
> Thank you for reviewing this paper. Could you read the authors' response and submit your final recommendation at your earliest convenience? Thank you.
>
> Best wishes,
> AE

---

### Review · Reviewer_ST3p · 2025-11-16

**Summary Of Contributions:**

This paper proposes a new _perception-coherent_ feature transfer objective for knowledge distillation, based on aligning the rank order of pairwise similarities between teacher and student embeddings. The key idea is that while absolute feature magnitudes and layer geometries may differ across architectures, the teacher’s internal perception structure can still be transferred through a rank-based loss. The method is label-free (the student receives no ground-truth supervision) and is applicable even when teacher and student differ significantly in capacity, architecture, or embedding dimensionality.

The authors demonstrate two main empirical results: (1) strong feature transfer performance in representation retrieval tasks on CIFAR-10 and a subset of CUB-200, where the proposed loss outperforms KD, FitNet, PKT, and other distillation baselines; and (2) competitive or better classification results compared to standard distillation methods (KD, FitNet, VID, PKT, and CRD) under heterogeneous teacher-student settings on CIFAR-100, such as ResNet-50 → MobileNetV2 and ResNet-32×4 → ShuffleNet.

**Key Strengths:**

-   Clear and well-motivated formulation based on rank consistency.

-   Unsupervised student training setup (no labels) while still leveraging teacher knowledge.

-   Demonstrates flexibility under cross-architecture and dimension-mismatched distillation settings.

-   Strong gains over several established KD baselines in both feature retrieval and classification.

-   Ablations on batch size, temperature, and student capacity improve transparency.


**Key Weaknesses (high-level):**

-   Representation experiments use a very small and non-standard student model, limiting generality.

-   The retrieval setup aligns closely with the proposed loss, making it unclear how broadly the representational gains extend.

-   No controlled same-architecture KD setup (e.g., ResNet-50 → ResNet-18).

-   No compression ratio reporting, despite KD’s strong connection to model compression.

-   All experiments are on small datasets (CIFAR, CUB subset), raising scalability questions.

**Audience:**

Yes

**Audience Explanation:**

Researchers working on knowledge distillation, model compression, and feature transfer would find this paper relevant, especially given its cross-architecture applicability, and improved results over established KD baselines.

**Broader Impact Concerns:**

I did not see any broader impact concerns.

**Claims And Evidence:**

Yes

**Claims Explanation:**

The paper claims that its rank-based loss can transfer the teacher’s internal perception structure to a student without using labels, that this works even when the teacher and student differ in architecture and feature dimensionality, and that this leads to improved representation transfer and competitive knowledge distillation performance. These claims are supported by the experiments. The retrieval results on CIFAR-10 and a CUB-200 subset show clear improvements over KD, FitNet, PKT, and HKD under identical teacher–student setups. The CIFAR-100 classification results demonstrate that the method matches or exceeds established KD methods, including CRD, across heterogeneous teacher–student pairs such as ResNet-50 to MobileNetV2 and ResNet-32×4 to ShuffleNet. The ablations on batch size, temperature, and student capacity further support the proposed design. While the evidence is convincing within this scope, several aspects of the evaluation could be strengthened, and I outline these in the requested changes section.

**Requested Changes:**

The following experiments are requested.

1.  **Controlled KD setup:**
    While the heterogeneous teacher–student pairs are interesting, a standard homogeneous setting (e.g., ResNet-50 → ResNet-18) would serve as a clean baseline to isolate the effect of the proposed loss from architectural mismatch. These should also be compared to standard distillation methods (KD, and CRD).

2.  **Representation claim clarity:**
    The retrieval experiments show strong improvements over KD-style baselines, but the setup is non-standard and tightly aligned with the proposed ranking loss. If the authors intend to make claims about representation learning more broadly, a comparison to at least one SSL baseline (e.g., DINO) or a cross-dataset transfer probe would strengthen the argument. Otherwise the claim should be framed explicitly within the KD context.

3.  **Compression ratio reporting:**
    Given that KD is fundamentally a model compression technique, it would be helpful to report teacher vs. student parameter counts or FLOP ratios, so readers can better interpret the practical efficiency gains. Since CRD is one of the main KD baselines and is known to report such compression metrics, including similar comparisons here would make the results easier to contextualize.

---

> ### Author Response · Authors · 2025-11-30
> **Response by authors**
>
> First of all, the author would like to thank the reviewer for the thoughtful remarks. Many aspects of the discussion will be incorporated into the revised paper.
>
> ## Homogeneous setting
> > While the heterogeneous teacher–student pairs are interesting, a standard homogeneous setting (e.g., ResNet-50 → ResNet-18) would serve as a clean baseline to isolate the effect of the proposed loss from architectural mismatch.
>
> Thank you for the constructive feedback. We appreciate the suggestion to include a homogeneous teacher–student baseline (e.g., ResNet‑50 → ResNet‑18). However, we would like to clarify our motivation and focus:
>
> - Our work is intentionally designed for heterogeneous teacher–student settings (i.e., different architectures), because this scenario reflects a realistic and widely‑needed demand: often in practice one wants to transfer knowledge from a large, possibly very different, model (teacher) to a small or efficient model (student) that may have a substantially different architecture. In such cases, traditional KD assumptions (same or very similar architecture) do not hold.
>
> - We believe that in homogeneous settings — where teacher and student share essentially the same architecture or processing structure — standard KD methods are already sufficiently effective, and the relative gain from a specialized loss (like ours) may be  redundant.
>
> That said,  we believe it is more appropriate to frame our claims within this context, rather than extending them to homogeneous settings where simpler baselines suffice. We agree that we should clearly state in the text that our focus is on heterogeneous teacher–student distillation, and mention explicitly that we do not aim to replace or improve KD in homogeneous settings. We will update the manuscript accordingly to make this scope explicit.
>
> ## Representation claim clarity
>
> Thank you for your insightful comment. We would like to clarify that our work focuses specifically on a new measure (perception coherence) for  **feature representation transfer between a teacher and a student model**, rather than general representation learning. Consequently, our comparisons are framed within the knowledge distillation (KD) context. While we agree that broader self-supervised learning baselines (e.g., DINO) are interesting, they fall outside the intended scope of our study. We will clarify this explicitly in the text to ensure that our claims remain properly contextualized within the KD framework.
>
> ## Compression ratio reporting
>
> Thank you for this recommendation. Here are the result of computational cost comparison.
>
> **Note:**
> - **MAC** = Multiply–Accumulate
> - **1 MAC ≈ 2 FLOPs** (one multiply + one add)
>
> |                | CIFAR10     | SVHN       | CIFAR100 (ResNet-50 → MobileNetV2) | CIFAR100 (ResNet-32x4 → ShuffleNetV1) | CIFAR100 (ResNet-32x4 → ShuffleNetV2) |
> |----------------|-------------|------------|-------------------------------------|-----------------------------------------|-----------------------------------------|
> | **Teacher model** | 557.22 MMac | 1.82 GMac  | 1.31 GMac                          | 1.09 GMac                               | 1.09 GMac                               |
> | **Student model** | 523.15 KMac | 37.26 MMac | 7.35 MMac                          | 41.58 MMac                              | 46.06 MMac                              |
>
> Kind regards,
>
> The authors

---

### Decision · Action_Editor_ezqL · 2026-02-05

**Recommendation:** Accept with minor revision

**Additional Comments:**

The authors didn't make the rebuttal visible to two of the reviewers. As such, two reviewers recommended rejection. After EIC making the rebuttal visible, unfortunately the reviewers didn't have a chance to check it. As a result, I reviewed the author rebuttal and found that the concerns were addressed well in the rebuttal.

I recommend acceptance of this paper, and I suggest the authors revise the manuscript according to reviewers' suggestions. In particular, please add the new experimental results in the final version.

**Audience:**

Yes

**Audience Explanation:**

The researchers and engineers working on training or deployment of models on the edge will be interested in this work.

**Claims And Evidence:**

Yes

**Claims Explanation:**

This paper proposes a perception-coherent feature transfer objective for knowledge distillation that aligns the rank order of pairwise similarities between teacher and student embeddings. The key insight is that, although absolute feature magnitudes and layer geometries may differ across architectures, the teacher’s internal perception structure can still be effectively transferred via a rank-based loss.

The experimental results, including the additional experiments required by reviewers, demonstrate the state-of-the-art performance of the proposed method.